

# Super-high-resolution aerial imagery datasets of permafrost landscapes in Alaska and northwestern Canada

Tabea Rettelbach[1,2,*], Ingmar Nitze[1,*], Inge Grünberg[1], Jennika Hammar[1], Simon Schäffler[1,3], Daniel Hein[4], Matthias Gessner[4], Tilman Bucher[4], Jörg Brauchle[4], Jörg Hartmann[5], Torsten Sachs[6], Julia Boike[1,7], and Guido Grosse[1,2]

[1]Alfred Wegener Institute Helmholtz Centre for Polar and Marine Research, Permafrost Research Section. Telegrafenberg A45, 14473 Potsdam, Germany.
[2]Potsdam University, Institute for Geosciences. Karl-Liebknecht-Str. 18, 14478 Potsdam, Germany.
[3]Ludwig-Maximilians-Universität München, Department of Geography. Luisenstr. 37, 80333 München, Germany.
[4]German Aerospace Center, Institute of Optical Sensor Systems. Rutherfordstr. 2, 12489 Berlin, Germany.
[5]Alfred Wegener Institute Helmholtz Centre for Polar and Marine Research, Atmospheric Physics Research Section, Klußmannstr. 3d, 27570 Bremerhaven, Germany.
[6]Deutsches GeoForschungsZentrum, Telegrafenberg, 14473 Potsdam, Germany
[7]Humboldt-Universität zu Berlin, Geography Department. Rudower Chaussee 16, 12489 Berlin, Germany.
[*]These authors contributed equally to this work.

**Correspondence:** Guido Grosse (guido.grosse@awi.de)

**Abstract.** Permafrost landscapes across the Arctic are very susceptible to a warming climate and are currently experiencing rapid change. High-resolution remote sensing datasets present a valuable source of information to better analyze and quantify current permafrost landscape characteristics and impacts of climate change on the environment. In particular, aerial datasets can provide further understanding of permafrost landscapes in transition due to local and widespread thaw. We here present a new dataset of super-high-resolution digital orthophotos, photogrammetric point clouds, and digital surface models that we acquired over permafrost landscapes in northwestern Canada, northern, and western Alaska. The imagery was collected with the Modular Aerial Camera System (MACS) during aerial campaigns conducted by the Alfred Wegener Institute in the summers of 2018, 2019, and 2021. The MACS was specifically developed by the German Aerospace Center (DLR) for operation under challenging light conditions in polar environments. It features cameras in the optical and the near-infrared wavelengths with up to 16 megapixels. We processed the images to four-band (blue - green - red - near-infrared) orthomosaics, digital surface models with spatial resolutions of 7 to 20 cm, and 3D point clouds with point densities up to 44 pts/m$^3$. This super-high-resolution dataset provides opportunities for generating detailed training datasets of permafrost landform inventories, a baseline for change detection for thermokarst and thermo-erosion processes, and upscaling of field measurements to lower-resolution satellite observations.

## 1 Introduction

In a globally warming world, detailed monitoring of the regions affected by rapid climate change is one key to understanding the underlying ecosystem and landscape processes and quantifying their consequences to the environment (Bartsch et al., 2021).





The cryosphere experiences some of the most profound impacts of climate warming, as it is especially vulnerable to increasing temperatures and, in turn, its disappearance further accelerates environmental change. It is currently estimated that the Arctic is warming two to four times as fast as the rest of the globe, a phenomenon largely driven by the so-called Arctic amplification through the sea-ice-albedo feedback (Walsh, 2014; Jansen, 2020; Richter-Menge and Druckenmiller, 2020; AMAP, 2021; Yu et al., 2021; Rantanen et al., 2022). Permafrost is one of the most important components of the cryosphere. Any ground with temperatures <0 °C for a minimum of two consecutive years is considered permafrost, which therefore can cover the range of very ice-rich soils to non-porous bedrock. According to the most recent study, permafrost is found in approximately 14 to 16 million km$^2$, or 15 %, of the exposed land surface area in the Northern Hemisphere (Obu, 2021). Atmospheric warming causes significant land surface warming and precipitation changes to these extensive landscapes, which result in enhanced permafrost thaw (Guo et al., 2020; Vincent et al., 2017; Jorgenson and Grosse, 2016; Grosse et al., 2016). Melt of ground ice in ice-rich permafrost significantly impacts landscape topography, geomorphology, pedology, hydrology, and vegetation structure and distribution. With these factors, the entire ecology of permafrost ecosystems is affected. In addition, anthropogenic infrastructure built on and into permafrost like roads, houses, and pipelines is increasingly at risk, maintenance costs are rising strongly, and damages have led to loss of economic value and increased risk of environmental hazards in recent years, strongly affecting the lives and livelihoods of the communities in the permafrost region (e.g., Hjort et al., 2022; Bartsch et al., 2021; Larsen et al., 2021; Miner et al., 2021). The complex interplay between these changing environmental factors and both gradual or abrupt permafrost thaw results in a broad range of ecological and economical impacts and consequences that manifest very differently across spatial scales. Accordingly, the response of permafrost landscapes in the face of climate change requires careful observation and monitoring, and remote sensing offers excellent tools and methods for this across large and remote regions. Remote sensing can deliver spatially continuous and comprehensive insights for land surface conditions across large areas affected by permafrost thaw throughout the Arctic (Jorgenson and Grosse, 2016). Per definition, permafrost is a subsurface phenomenon, but the consequences of soil thaw, ground-ice melt, and surface subsidence and erosion result in characteristic surface processes and landforms that can be monitored from above. Earth observation imagery thus provides both a basis for spatially comprehensive permafrost landscape monitoring through remote sensing image analysis and a database of observations that can be consulted for purposes of validating in-situ field measurements or predictions from modeling approaches. Permafrost phenomena that can be directly observed with remote sensing include thaw slumping (e.g., Runge et al., 2022; Bernhard et al., 2022; Lewkowicz and Way, 2019), coastal erosion (e.g., Wang et al., 2022; Jones et al., 2018; Irrgang et al., 2018), ice-wedge degradation (e.g., Frost et al., 2018; Rettelbach et al., 2022; Jorgenson et al., 2022), growing and draining thermokarst lakes (e.g., Nitze et al., 2020; Lara et al., 2021; Jones et al., 2020), ground subsidence (e.g., Zwieback and Meyer, 2021; De la Barreda-Bautista et al., 2022; Liu et al., 2014), and post-fire thermokarst dynamics (e.g., Jones et al., 2015; Rettelbach et al., 2021; Iwahana et al., 2016). Remote sensing also allows monitoring of further ecological processes like beaver-damming activities (e.g., Jones et al., 2021; Tape et al., 2022, 2018), and anthropogenic activities such as road construction and maintenance (e.g., Raynolds et al., 2014; Walker et al., 2015; Kaiser et al., 2022) which affect permafrost directly or indirectly. Airborne remote sensing plays an essential role in bridging the gap between hyper-local, in-situ field studies and close-range remote sensing with uncrewed aerial systems (UAS) or ground-based remote sensors at one end, and satellite-based remote sensing

covering very large spatial areas at lower spatial resolutions at the other end. Airborne sensors offer high operating flexibility and can cover significantly larger areas than UAS (Oldenborger et al., 2022; Boike and Yoshikawa, 2003), while also capturing great spatial detail. Recent aerial campaigns within the NASA ABoVE project focused on acquiring hyperspectral, SAR, and laser altimetry data over permafrost regions in Alaska and NW Canada to observe wetlands, greenhouse gas (GHG) emissions, and active layer dynamics (Miller et al., 2019). In combination with historical aerial imagery or laser scanning datasets, modern optical airborne datasets have been used to quantify thaw subsidence following disturbances (Jones et al., 2013; Zhang et al., 2023), lake change (Jones et al., 2012), broad landscapes changes (Jorgenson et al., 2018), coastal erosion (Jones et al., 2020; Obu et al., 2017), ice-wedge degradation (Liljedahl et al., 2016; Rettelbach et al., 2021), and vegetation dynamics (Tape et al., 2006).

We here report on a new airborne image dataset that we collected across extensive permafrost-affected areas and broad environmental gradients in Alaska and Northwest Canada in 2018, 2019, and 2021. The data includes super-high-resolution multispectral images in the visible (red-green-blue, RGB) and near-infrared (NIR) wavelengths, captured with the advanced Modular Aerial Camera System (MACS) developed by the German Aerospace Center (DLR), and flown onboard the polar planes of the Alfred Wegener Institute Helmholtz Centre for Polar and Marine Research (AWI). From the densely overlapping imagery, we derived super-high-resolution (up to 7 cm/px) RGB and NIR ortho-image mosaics and dense photogrammetric point clouds as well as digital surface models (DSMs). Here we describe these datasets of derived image products, which are archived and accessible on the PANGAEA scientific data repository (Rettelbach et al., 2023).

# 2   Data acquisition

The super-high-resolution aerial image datasets published here were acquired during three AWI airborne campaigns conducted over permafrost regions in northwestern North America in the summers of 2018, 2019, and 2021. All surveys were flown with the AWI polar planes Polar-5 or Polar-6 using the MACS.

## 2.1   Study areas

The study regions in northwestern North America are characterized by extensive permafrost landscapes with gradients from sporadic to discontinuous to continuous permafrost extent. In the North, tundra ecosystems are predominant, while farther south the ecosystem transitions to shrub tundra and boreal forests. All study areas cover both coastal as well as more continental areas and feature Köppen-Geiger climates ranging from Dfc (cold, no dry season, cold summer), to Dsc (cold, dry summer, cold summer), to ET (polar, tundra) (Beck et al., 2018).

### 2.1.1   Yukon and Northwest Territories, Canada - 2018

From 15 to 29 August 2018, we surveyed several transects in both Yukon and the Northwest Territories, Canada, covering a total area of 746 km$^2$ and ranging from the Mackenzie (Deh-Cho/Kuukpak) Delta in the East to Herschel Island (Qikiqtaruk) in the Canadian Beaufort Sea close to the Alaskan border in the West and the village of Fort MacPherson (Teetł'it Zheh) in the



**Figure 1.** Footprints of the acquired aerial imagery for the study regions of (a) North Alaska, (b) Northwest Canada, and (c) West Alaska. (d) Location of the three study regions in northwestern North America. Black areas show the footprints of all available imagery, while yellow areas cover the footprints of the here published and presented datasets. Basemaps in (a), (b), and (c): © 2015 Google. Basemap in (d): Esri, HERE, Garmin, © OpenStreetMap contributors, and the GIS user community.

South. The survey mainly included the corridor of the Inuvik to Tuktoyaktuk Highway (ITH) and the Trail Valley Creek (TVC)
research watershed (Fig. 1b, d). ITH is a 137 km long gravel highway which was officially opened in 2017. TVC is located
about 55 km northeast of Inuvik and has been in the focus for research on snow, permafrost, vegetation, and hydrology since
1991 (Marsh et al., 2010; Antonova et al., 2019; Wilcox et al., 2019; Grünberg et al., 2020). The sites are characterized by
medium ice-rich permafrost and are located within the continuous permafrost zone (Obu et al., 2019). Thermokarst lakes and
degrading ice-wedge landscapes are abundant throughout the Mackenzie Delta, in the inland, and especially pronounced also




along the ITH. Along the coast of Herschel Island and the Canadian Beaufort Sea, permafrost coastal erosion is an important
process (Lantuit and Pollard, 2008) and some coastal segments are affected by strong retrogressive thaw slump activity (Lantuit
and Pollard, 2005). The coastal landscapes and hinterlands features terrain rich in lakes and drained lake basins, hummocky and
rolling terrain with predominantly ground moraines (fine-grained and stony tills) with some interspersed alluvial, glaciofluvial
and lacustrine deposits (Duk-Rodkin and Lemmen, 2000). The highway crosses the tundra-taiga ecotone and the vegetation
changes from dwarf-shrub tundra in the north to open-canopy spruce forests in the south (Timoney et al., 1992). TVC is located
at the northern edge of the treeline zone and is characterized by a mix of dwarf shrubs and herbaceous tundra, upright shrub
tundra, and open spruce woodlands. The ITH corridor is characterized by a significant climatic gradient, with coastal conditions
drier and colder than the inland (Burn and Kokelj, 2009). The mean annual air temperatures 1990-2020 were -7.1 and -8.9 °C
at Inuvik and Tuktoyaktuk, respectively (Government of Canada, 2022). In the weeks prior to the image acquisitions, no
precipitation was recorded at the Inuvik (Mike Zubko) or the Aklavik Airport Stations. For the later flights at the end of August
(i.e., ITH on 29 August 2018), foliage has already transitioned towards fall colors, which is visible in the imagery.

### 2.1.2 Alaska North Slope, USA - 2019

The 2019 airborne campaign with Polar-6 focused on the Alaska North Slope (Siḷaliñiq), USA. Main targets in this study region
were coastal segments between Utqiaġvik (Barrow) and Pt. McLeod, the Outer Arctic Coastal Plain north of Teshekpuk Lake,
the Ikpikpuk Delta, historic fire scars between the Inner Arctic Coastal Plain, the Ikpikpuk Sand Sea, the foothills north of the
Brooks Range, and the Anaktuvuk (Anaqtuuvak) River fire scar east of Umiat (Fig. 1a, d). This campaign was conducted from
13 to 31 July in 2019 and covered 1766 km$^2$ in total. This region is located entirely within the zone of continuous permafrost
extent (> 90 % permafrost coverage) and mean annual ground temperatures range from -12 to -5 °C (Obu et al., 2019; Jorgen-
son, 2008). Thermokarst lakes and drained thermokarst lake basins are the most prevalent permafrost features in the northern
lowland landscapes of the studied region but many other types of thaw landforms are found as well and indicate widespread
presence of ice-rich ground (Farquharson et al., 2016). Along the Alaskan Beaufort Sea coastline, strong permafrost coastal
erosion is occurring in many segments with highest rates reported from Drew Point (Jones et al., 2020). Further south, terrain
becomes more sloping towards the Brooks Range foothills and barely any lakes are present. The vegetation is characterized by
a mossy tundra with sedges and dwarf-shrubs in the North, with shrub sizes increasing along the southward gradient (Raynolds
et al., 2019). Mean annual air temperatures in Utqiaġvik were -10.1 °C with a mean annual precipitation of 137 mm (1991-
2020) (NOAA, 2023), which is highly influenced by its coastal location. Precipitation occurs predominantly during the summer
and fall months. Cloud cover and fog are persistent in coastal areas, creating challenging conditions for (imaging) flights. In
the four weeks prior to the flight campaign, average precipitation (40 mm) was recorded (NOAA, 2023). During the campaign,
precipitation increased slightly and records from Utqiaġvik measured further 34 mm of rain during these acquisition dates. By
the time the data was collected, snowmelt for the 2019 season had already concluded. During the observation period, weather
was variable from cloud-free to overcast and rain. On several sunny days across large parts of the target area (North Slope),
acquisition flights could not be flown, as fog at the aircraft's base in Utqiaġvik hindered takeoff, missing optimal acquisition
conditions.



### 2.1.3 Northwest Alaska, USA - 2021

In 2021, the aerial campaign was conducted between 25 June and 10 July and covered a total area of 3591 km$^2$ in Northwest Alaska. Main targets of this campaign were coastal segments along the northern and central Baldwin Peninsula, the outer areas of the Noatak and the Kobuk (Kuuvak) deltas, the lake- and basin-rich Cape Espenberg lowlands, the Nome-Taylor Highway (Kougarok Road) and the Imuruk volcanic field on the Central Seward Peninsula, historic fire scars near Buckland and in the lower Noatak Valley, the Selawik thaw slump, various known beaver-affected sites in the region, as well as the eight vil-
lages of Buckland (Kaniq), Deering (Ipnatchiaq), Kivalina (Kivalliñiq), Kobuk (Laugviik), Kotzebue (Qikiqtaġruk), Selawik (Akuliġaq), Shishmaref (Qikiqtaq), and Shungnak (Nuurviuraq) (Fig. 1c, d). The majority of these areas are located in a region, where continuous permafrost transitions to the discontinuous permafrost zone (50 to 90 % permafrost coverage) with mean annual ground temperatures ranging from 0 to -4 °C (Jorgenson, 2008; Obu et al., 2019). Some areas on the Northern Seward Peninsula and in the Noatak Valley are already part of the continuous permafrost zone. The region is characterized
by a wide range of permafrost landscape dynamics. Most of the target sites have been affected by thermokarst lake expansion and drainage events in the recent past (Jones et al., 2012; Nitze et al., 2020) or lake (re)formation by beaver activity (Tape et al., 2018, 2022; Jones et al., 2021); coasts are partially affected by retrogressive thaw slumps and coastal erosion (Farquharson et al., 2018); widespread subsidence and thermokarst activity can be found throughout the surveyed areas. On the Northern Seward Peninsula, the landscape is dominated by wetland tundra and thermokarst lakes, with sparse vegetation. The
Central Seward Peninsula is however characterized by low to tall shrubs in between otherwise mainly moist tundra landscapes (Raynolds et al., 2002). The surveyed areas at the western foothills of the Brooks Range show tundra, mainly dominated by graminoids and dwarf and tall shrubs (Ueyama et al., 2013). In the surveyed areas in the east, we find a mosaic of varying local conditions from wetland to boreal forests. Mean annual air temperatures range from -7.6 to -1.2 °C, average annual precipitation at the Kotzebue weather station is around 289 mm. While the average month of June is rather dry in Kotzebue with
mean monthly precipitation of 15 mm, July ushers in the wet season with mean monthly values of 41 mm (1991-2020) (NOAA, 2023). In 2021, however, the area experienced record-breaking 137.9 mm of rainfall in July alone (NOAA, 2023). Given the timing of the aerial imaging campaign, this might have impacted the properties of water bodies and the hydrological state of the landscape in the acquired datasets.

## 2.2 Multispectral sensor

For all three campaigns, we used the custom-built Modular Aerial Camera System (configuration: MACS-Polar18) developed by the Institute of Optical Sensor Systems of the DLR. The general concept of MACS is described by Lehmann et al. (2011). It was specifically adapted for handling challenging light conditions in Arctic environments, where very dark (water, dark bare soil) and bright (snow, ice) surfaces often co-exist in target areas (Brauchle et al., 2015). The MACS consists of a computing unit and a sensor unit. The computing unit comprises sub-assemblies including an external interface box, an auxbox, and the
main computer. The sensor unit contains an inertial measurement unit (IMU) and three 16 megapixel (MP) cameras: one nadir-looking NIR camera, and two visible RGB cameras with overlapping right- and left-looking (+/- 8.5°) view directions (Fig. 2a,





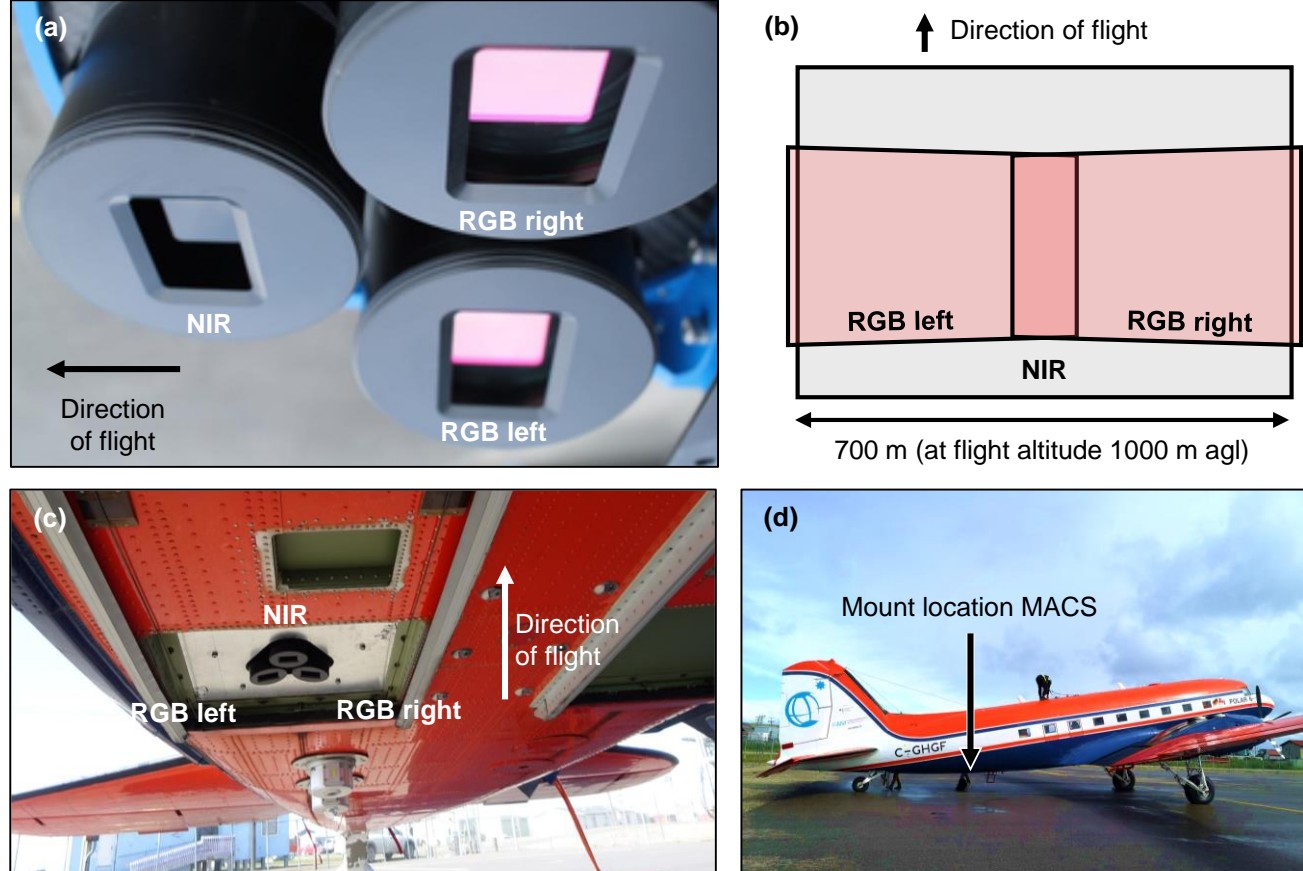

**Figure 2.** (a) MACS sensor heads installed in the airplanes' belly ports. The left sensor is the NIR camera, the two right sensors the RGB cameras. (b) Ground footprints of the three cameras at a flight altitude of 1000 m agl, showing the overlapping ground areas. (c) and (d) Mount location of the camera within the Basler-BT65 aircraft.

b). Technical specifications on both sensors are summarized in Table 1. The maximum frame rate is 4 frames/s. The IMU root mean square (RMS) accuracy is 0.006°. The MACS-Polar18 was operated onboard the AWI Polar-5 (2018) and the Polar-6 (2019, 2021) polar research aircrafts, which are Douglas DC-3 planes refitted to Basler BT-67 planes for operation in harsh

polar environments, with the sensor unit installed in a belly port of the planes (Fig. 2c and d).

## 2.3 Survey design

Our surveys were guided by multiple research questions associated with permafrost landscape and ecosystem change. Accordingly, our observation targets included permafrost landscapes affected by thaw, eroding coasts, hydrological and vegetation characteristics, as well as settlements and infrastructure threatened by damage from permafrost thaw. We prioritized sites ac-

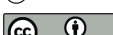

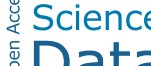

**Table 1.** Technical specifications of the RGB and NIR sensors of the MACS-Polar18 configuration.

| | RGB sensor | NIR sensor |
|---|---|---|
| **wavelength range** (FWHM*) | B: 450-510 nm<br>G: 490-580 nm<br>R: 580-700 nm | 715-950 nm |
| **sensor resolution** | 16 MP | 16 MP |
| **IFOV**† | 81 $\mu$rad | 140 $\mu$rad |
| **FOV**‡ (cross × along track) | 23° × 16° | 40° × 22° |
| **GSD**§ (at 1000 m agl) | 8 cm | 15 cm |

\* full width at half maximum; † instantaneous field of view; ‡ field of view; § ground sampling distance

cording to a) areas where ground sampling had been performed during previous field campaigns to allow for linking local-scale observations to regional-scale airborne data; b) footprints of previous aerial campaigns with available published datasets (i.e., Marsh et al., 2010; Miller et al., 2019) to provide data for change detection analyses and time-series observations; and c) areas with infrastructure to assess permafrost thaw-associated vulnerabilities (e.g., Van der Sluijs et al., 2018) and contribute to monitoring and mitigation efforts of other institutions or local authorities. For the surveys, we flew either elaborate grids

or single-track transects for different areas of interest depending on the specific observation goals for the respective sites. The size and shape of a target area also had an influence on the survey design. The flight altitude, and correlated with this, the highest possible ground sampling distance (GSD), was similarly chosen with the prospective research question in mind. Given the resolution of 16 MP of the MACS, and the majority of flights conducted at altitudes between 500 m and 1500 m agl, the GSDs of the acquired imagery range between 4.4 and 12.5 cm/px for the RGB sensors and between 7.7 and 21.5 cm/px for the

NIR sensor. The along-track overlap between single image captures is 80 % for all datasets. For targets flown in grid-mode, the across-track overlap ranges from 28 % (2018), to 45 % (2018, 2019) up to 60 % (2021). For the photogrammetric processing, this leads to every ground location being captured by 10 to 15 images, and thus from 10 to 15 slightly different viewing angles. This number declines towards the edges of grids and can further vary with deviant angles of roll, pitch, and yaw of the aircraft induced by internal (i.e., pilot) or external (i.e., wind, drift) influences.

While we here only publish datasets generated from targets that we flew in grid-patterns which were suitable for photogrammetric processing (labeled in yellow in Fig. 1a-c), we also did capture additional imagery on single-track transect flights (labeled in black in Fig. 1; master tracks available via Hartmann (2018) and Grosse et al. (2019, 2021)). Raw, individual images from additional flight tracks (see black areas in Fig. 1), not covered by the processed data described in this publication is available upon demand to the authors until further processing and public data archival has been conducted.



# 3 Data processing

We processed all data based on the workflow shown in Fig. 3. Multiple software were used in the workflow, but most data handling for pre- and post-processing is implemented automatically through Python scripts (`01_SetupData.py` and `02_Postprocessing.py`, both available via https://github.com/awi-response/macs_processing/; see Sec. 7). Software, such as DLR MACS Image-Pre-Processor (MIPPS) and WhiteboxTools is also operated via Python. The main structure-from-motion (SfM) processing is handled with the GUI-based Pix4Dmapper (version 4.6.4). The following sections describe the substeps of the processing in more detail.

## 3.1 Pre-processing of raw MACS images

Upon acquisition, the raw Bayer-pattern images (Bayer, 1976) are stored in the DLR-proprietary .macs-format. These files contain information on the acquisition time in UTC; the image's geospatial properties, including the coordinates of the image center and the altitude of flight above WGS84 ellipsoid (measured by the onboard GNSS receiver); the attitude of the camera at capture time with values for roll, pitch, and yaw (measured by the IMU); as well as the information on which sensor captured the respective image. In this format, the files can be viewed in MACS-Viewer software and initially processed within the Mosaica application, both available within the MACS-Box, a Windows-based software package for MACS flight planning, raw data view, data preprocessing, and data export developed by DLR (DLR, 2019a). To create the desired mosaics with Pix4Dmapper and to share the data in a more commonly used format, we applied pre-processing and cleaning operations and exported these georeferenced files to tagged image file format (TIFF, TIF-format). The pre-processing is done using the MIPPS; another tool within the MACS-Box. The pre-processing steps are described in the following.

### 3.1.1 Devignetting and file format conversion

The acquired raw images are affected by a vignetting effect, which is characterized by a decrease in illumination from the center towards the edges and corners of each image. To overcome this systematic effect, we applied a devignetting algorithm (DLR, 2019b). It requires the beforehand calibrated camera's Dark Signal Non Uniformity (DSNU) parameters as input and corrects the brightness to create a more homogeneous lighting. Fig. 4 shows the difference between an exemplary image before (Fig. 4a) and after devignetting (Fig. 4b). Furthermore, we transformed each image file from the proprietary MACS image format to 16-bit TIFFs without projection. For this, we decompressed the original images and applied a demosaicking algorithm from one-channel Bayer-pattern to three-channel RGB, resulting in an image size of 30 MB (NIR) and 90 MB (RGB). Metainformation on the camera's position and attitude was stored in an external text file (nav-file), and formatted to match the requirements for processing with Pix4Dmapper. Both the devignetting and the format conversion were carried out with MIPPS, the nav-file was generated with a custom Python script. The camera definition file for devignetting and the MIPPS configuration file are available via git repository (at https://github.com/awi-response/macs_processing/blob/main/pix4D_processing_templates/pix4d_CameraDef_MACS_Polar18.xml and https://github.com/awi-response/macs_processing/





**Figure 3.** Workflow diagram of the processing steps to generate the photogrammetric orthomosaics, point clouds, and digital surface models from the multispectral MACS images. The diagram also provides an overview of additional by-products published with this manuscript.

tree/main/mipps_scripts respectively; see Sec. 7). The nav-files are specific to each project and are thus distributed via the respective PANGAEA dataset.





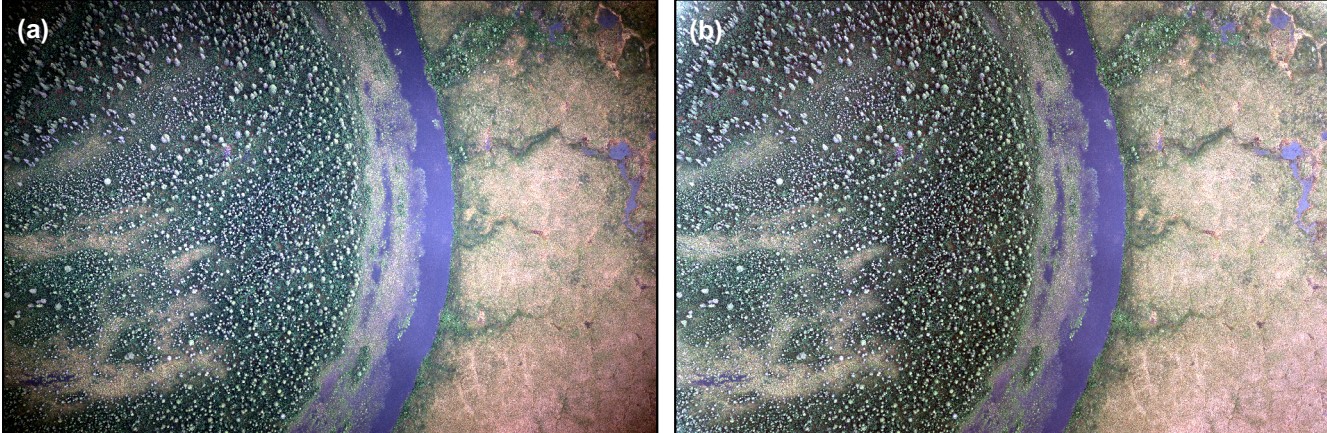

**Figure 4.** (a) Original MACS image shows the intensity of the vignette towards the borders of the image. (b) Same image after devignetting with pre-processing MIPPS script. The contrast of light intensities between the image center and the image borders is decreased, thus providing a more homogenous image as input for mosaicking. Image location: Oxbow lake from Selawik River at 66.4662° N, 157.8018° W. Image taken on 1 July 2021.

### 3.1.2 Radiometric scaling for matching RGB and NIR bands

In order to combine all four bands of the blue, green, red, and near-infrared wavelengths in the output mosaics the images
from both the RGB and NIR cameras need to be combined. For radiometric matching, we applied a linear scaling factor to the RGB acquisitions. We determined the scaling factor from the difference in shutter timing between the two sensors. During some of the flights, such differing shutter timings were chosen, to accomodate for the specific environmental conditions, such as brightness levels of the ground features, or expected shifts in illumination throughout the survey time. Subsequently, we scaled the data to exploit the entire 16-bit information range. This step also results in a more homogenized output among the
three campaigns and target sites that were flown under varying illumination conditions and with slightly different initial camera parameter settings. It also simplified the debugging and process control.

### 3.1.3 File preparation

To simplify the process of mosaicking the images with Pix4Dmapper in a later step (see Sec. 3.2), we added information of the acquiring sensor type to each image, which allowed Pix4DMapper to automatically assign the correct camera definition. This
was done via exif tags which then contained the parameter 'right', 'left', or 'center' for the right or left RGB sensor or the NIR sensor, respectively.





## 3.2 Structure-from-motion orthomosaic and point cloud generation

We processed the individual georeferenced images with the photogrammetry software Pix4Dmapper of the Pix4D imaging tool suite. All processing was done on a computer with an AMD EPYC 7702P 64-core processor running at 2 Hz and 512 GB
of available RAM, running Microsoft Windows 10. Computation was partly run on two NVIDIA A40 GPUs. For all target datasets, we created multispectral orthomosaics exported as GeoTIFFs and dense points clouds as LAS-format both for the RGB sensors and the NIR sensor. Orthorectification of the image mosaic is automatically carried out in Pix4Dmapper with the internally calculated elevation information (Pix4D, 2021a). We later processed the point clouds to DSMs (see Sec. 3.3). For large target areas exceeding 10.000 images, we created subprojects, so that Pix4Dmapper and the processing hardware could
manage computation. Geolocation information for each individual image, available as latitude and longitude coordinates for the image center point, the flight altitude (WGS84), and camera angles (Omega, Phi, Kappa), were provided in a separate nav-file in a Pix4D-readable format. For the photogrammetric orthomosaic generation, the software used this accurate geolocation information to identify neighboring images for tie-point matching and georeferencing. Pix4Dmapper largely accounts for exposure mismatches through a global offset and individual gain values per image per sensor type. With this global balancing,
it can correct or normalize brightness levels in images. During the photogrammetric calculation of the 3D point cloud, erroneous points can arise from e.g., noise in the original imagery. To ensure that these errors were not propagated into any subsequently created raster DSM, we filtered such outlying points (outliers in the z-dimension) by correcting their altitude with the median of their neighboring points (Pix4D, 2021a). Depending on the flight altitude of the carrier aircraft and thus the maximum GSD of the imagery, we processed the orthomosaics at either 7 cm (ca. 500 m agl), 10 cm (ca. 1000 m agl), or 20 cm (ca. 1500 m agl)
spatial resolution. For some datasets, slightly higher resolutions could be achieved, however, we decided to rather provide orthomosaics with homogeneous properties. The average point cloud densities range from 2.57 to 48.03 px/m$^3$ in the RGB and 1.18 and 17.85 px/m$^3$ in the NIR point clouds, depending on flight parameters. All steps were performed within Pix4Dmapper. For reproducing the analysis, we provide the Pix4Dmapper processing templates for each target GSD; they can be accessed via git repository (at https://github.com/awi-response/macs_processing/tree/main/pix4D_processing_templates; see Sec. 7).

## 3.3 Digital surface model calculation

To create a DSM from the processed RGB and NIR point clouds, we used WhiteboxTools (v.2.2.0, Lindsay et al., 2019) via the WhiteboxTools Python Frontend. DSMs can be generated from either the RGB-only point cloud, the NIR-only point cloud, or from combining both. We conducted a test of the three approaches and found that in most cases, using both the RGB and the NIR point cloud yielded best results. The majority of issues arose from pixels that were over- or undersaturated, which
made the matching more difficult and thus resulted in errors in the DSMs in those areas. A more detailed analysis can be found in Sec. A. Based on this finding, for each target area, we first merged all of the RGB and NIR point cloud tiles output by Pix4Dmapper to one point cloud. We then used inverse distance weighting (IDW) to calculate a continuous surface from the merged point cloud. From this initial surface model, we filled no-data gaps smaller than $5 * r - 1$, $r$ being the spatial resolution of the resulting dataset. This number is large enough to fill small water surfaces such as thermokarst trough ponds (and thus

avoid DSMs of polygonal terrain speckled by no-data gaps), but small enough that we do not need to make guesses for larger lakes. Subsequently, we applied surface smoothing in case some erroneous points potentially causing artifacts were missed (Lindsay et al., 2019). We then conducted a visual quality check of the final product: If any issues were visible, e.g., striping, bowling, imprecise edges, etc., we manually inspected the merged point cloud, determined from which individual point cloud (RGB or NIR) the error originated, and finally reprocessed the DSM from the higher quality point cloud only.

### 3.4    Post-processing

#### 3.4.1    Tiling

To be able to photogrammetrically process the large amounts of images on our end, and also provide datasets in sizes that can be handled by standard computing infrastructure and a wider range of post-processing software on the user side, we tiled the orthomosaics, point clouds, and DSMs. We first split the mosaicked data based on a grid with tiles of 5000 px by 5000 px

using the automatically created tile grid by Pix4Dmapper. The grid position is determined based on the north-west corner of an orthomosaic, with tile 1_1 covering the north-westernmost square of the target area. The first digit represents column number and the second digit represents row number. As no target dataset is exactly rectangular in shape, some tiles at the borders may also have non-rectangular shapes (see e.g., tile 1_3 in Fig. 5). Due to irregular shapes of some datasets, tile numbers are omitted if their column_row combination is not covered by data (i.e., tile 1_1 in the example dataset in Fig. 5). We provide geojson

files of the tiling pattern alongside our mosaics. The point clouds, which were initially tiled to another pattern by the software, were later re-tiled to match the tiling of the raster datasets. This operation was conducted with the las2las function of lastools.

#### 3.4.2    Image stacking

For each ortho-image tile, Pix4Dmapper generated two separate orthomosaics; a single-band NIR raster and an RGB raster with three bands. Both outputs have the same extent and spatial resolution, allowing us to stack them to a single four-band raster

with exactly superimposing pixels. Following band stacking, we reordered the multispectral bands so that the final output is ordered radiometrically: blue - green - red - near-infrared (R-G-B-NIR) following the typical band order of optical remote sensing datasets.

#### 3.4.3    Calculating pyramid layers and virtual rasters

To speed up display of the tiled and rather large datasets for end-users, we further computed GeoTIFF overviews accessible

as ovr-files, so-called pyramid layers. We computed pyramid layers for each ortho-image tile and DSM tile based on nearest-neighbor resampling and are now available at the following levels: 2, 4, 8, 16, 32, and 64. This step was performed via the gdal function gdaladdo. Finally, we created virtual raster tile (.vrt) mosaics using all tiles of a subset, to load all files at once without physical duplication of the data. This allows users to load imagery from an entire project in GIS or similar applications without the need to import all the individual tiles. We provide such virtual rasters for both the multispectral orthomosaic and

DSM data.





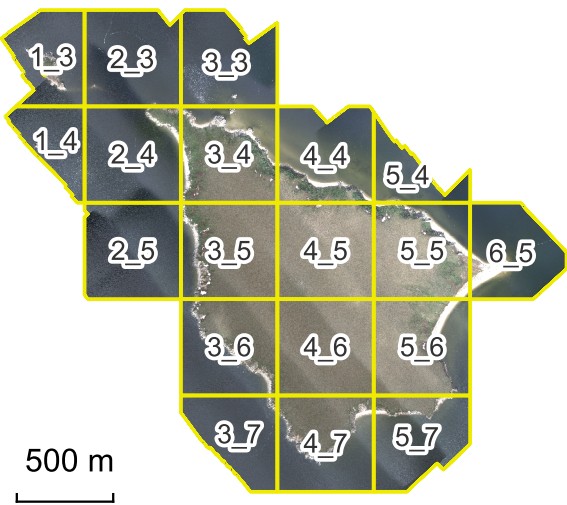

**Figure 5.** Tiling and tile naming scheme for processed orthomosaics, point clouds, and digital surface models displayed on dataset 'WA_ChamissoIsland_20210625_10cm_01'. Regular tiles measure 5000 px by 5000 px, with tiles at the border of the dataset forming an exception. Numbering follows a column_row pattern $i\_j$, with 1_1 being in the northern- and westernmost corner. $i$ increases eastward, while $j$ increases southward.

## 4 Data and metadata structure

We here publish three main datasets (1-3) and multiple further, supporting files (4a-g) for 102 areas of interest (AOI) of varying sizes. For each AOI the following files are available:

1. Multispectral orthophoto mosaic with the blue, green, red, and near-infrared band (as tiles);

2. Photogrammetrically-derived dense point clouds from both the NIR and the RGB sensors;

3. Digital surface model of the processed point cloud (as tiles);

4. Additional data:

    (a) Overview files (pyramid layers):

        i. For the multispectral orthophoto mosaics (as tiles);

ii. For the photogrammetric digital surface model (as tiles);

    (b) Virtual mosaic files:

        i. For the multispectral orthophoto overview tiles;

        ii. For the photogrammetric digital surface model overview tiles;





(c) Quick-look preview files (of the entire AOI)

    i. For an optical orthophoto mosaic;

    ii. For a color-infrared orthophoto mosaic;

    iii. For a digital surface model;

(d) Geojson-file of the tile footprints valid for the multispectral orthophoto mosaic, dense point cloud, and digital surface model;

(e) Pix4Dmapper automatically generated orthomosaicking quality report;

(f) Navigational text file (nav-file) with position and attitude camera information for each individual image used for the mosaics;

(g) Log-file of the steps performed during the automated data processing .

Additionally, we provide two superordinate GeoPackage (GPKG) files of data footprints. `aois.gpkg` combines the foot-
prints of all published datasets (Fig. 1, yellow area), and `image_footprints.gpkg` shows the footprints of all images taken during the three described campaigns (Fig. 1, black area, data available via PANGAEA, see Sec. 7). The latter also includes the areas that were covered by single swath transects and not flown in grid-mode, for example during transits between targets, and therefore were not processed photogrammetrically and not published alongside this manuscript. However, images of these additional areas will become available in future versions. The metadata on the campaign-level is available per project
dataset via PANGAEA. It is compatible with the Open Archives Initiative Protocol for Metadata Harvesting (OAI-PMH) and can thus be harvested in a variety of standardized formats (DC, ISO 19139, DIF, DataCite, internal PANGAEA XML schema) through automated tools (PANGAEA, 2023). Table 2 shows the list of all target areas with additional information on the day of acquisition, the area covered, the GSD of orthomosaics and the average point cloud densities.

## 4.1 Multispectral orthophoto mosaics and digital surface models

The resulting multispectral orthophoto mosaic for each target area is available in GeoTIFF as a four-band image (blue - green - red - near-infrared) (Rettelbach et al., 2023) and tiled into individual files of size 5000 px by 5000 px. DSMs are tiled to the exactly equivalent footprints. Depending on the flight altitude of the aircraft during the image acquisition, we achieved GSDs of up to 4.4 cm for the RGB sensors and 7.7 cm for the NIR sensor. As each dataset includes images from both sensor types, we have decided to select the approximate GSD of the NIR sensor, the lower-resolution of the two, as the common final
product pixel size. The vast majority of image surveys were usually conducted at one of three different main flight altitudes. Some selected targets (e.g., Cape Blossom in West Alaska in 2021) were covered twice or three times with different altitudes for comparative resolution tests. We have thus selected three different image resolutions for the final mosaicked products: For flights conducted at ca. 500 m agl, mosaics were processed at 7 cm/px; flights at ca. 1000 m agl were processed at 10 cm/px, and flights at altitudes ca. 1500 m agl resulted in mosaics of 20 cm/px GSD. All spatial datasets are projected to the UTM
WGS84 coordinate reference system (CRS). The UTM zones range from UTM 3N (Seward Peninsula, West Alaska) to 8N



**Table 2.** Overview of all published target areas with information on acquisition parameters, dataset coverage and resolution, and size of the datasets.

| Target name | region | Day of acquisition (YYYY-MM-DD) | Area covered (km$^2$) | Orthomosaic / DSM GSD (cm) | Average RGB PC density (points/m$^3$) | Average NIR PC density (points/m$^3$) | number of tiles | number of sub-projects |
|---|---|---|---|---|---|---|---|---|
| HerschelIslandEast | WC | 2018-08-15 | 1.38 | 7 | 48.03 | 17.85 | 15 | 1 |
| TukRoadGrid | WC | 2018-08-29 | 15.03 | 10 | 10.28 | 3.86 | 78 | 1 |
| TrailValleyCreek | WC | 2018-08-22 | 161.12 | 10 | 14.29 | 4.66 | 864 | 8 |
| CapeSimpson | NA | 2019-07-19 | 23.92 | 7 | 28.17 | 10.17 | 264 | 5 |
| AnaktuvukRiverFire | NA | 2019-07-22 | 34.94 | 7 | 23.94 | 8.32 | 391 | 5 |
| TeshekpukLakeNorth | NA | 2019-07-23 | 107.68 | 7 | 17.52 | 6.44 | 1043 | 11 |
| KetikFire | NA | 2019-07-27 | 81.35 | 7 | 36.27 | 12.23 | 819 | 9 |
| MeadeFire | NA | 2019-07-29 | 52.73 | 10 | 9.46 | 3.32 | 257 | 2 |
| NorthSlopeCentral | NA | 2019-07-29 | 58.56 | 10 | 8.82 | 3.08 | 309 | 4 |
| DrewPoint | NA | 2019-07-30 | 104.51 | 10 | 8.15 | 3.05 | 407 | 5 |
| IkpikpukDelta | NA | 2019-07-31 | 14.16 | 7 | 22.73 | 8.83 | 122 | 1 |
| ChamissoIsland | WA | 2021-06-25 | 7.44 | 10 | 24.44 | 8.36 | 19 | 1 |
| Kotzebue | WA | 2021-06-25 | 12.11 | 10 | 5.75 | 2.01 | 40 | 1 |
| CapeBlossom | WA | 2021-06-25 | 58.12 | 20 | 13.44 | 4.84 | 254 | 3 |
| BucklandFireScar | WA | 2021-06-27 | 50.82 | 7 | 36.87 | 12.73 | 515 | 6 |
| BaldwinPeninsulaNorth | WA | 2021-06-28 | 16.68 | 10 | 8.43 | 2.94 | 90 | 1 |
| Shishmaref | WA | 2021-06-28 | 9.52 | 10 | 11.32 | 4.27 | 45 | 1 |
| BPSouth | WA | 2021-06-28 | 85.67 | 20 | 4.20 | 1.51 | 116 | 3 |
| ShungnakKobukVillages | WA | 2021-07-01 | 20.87 | 10 | 5.91 | 2.45 | 120 | 1 |
| SelawikVillage | WA | 2021-07-01 | 5.74 | 10 | 5.81 | 2.35 | 31 | 1 |
| SelawikSlump | WA | 2021-07-01 | 15.67 | 10 | 5.76 | 2.34 | 77 | 1 |
| NoatakValleyN | WA | 2021-07-02 | 51.04 | 7 | 26.72 | 9.17 | 556 | 4 |
| NoatakValleyS | WA | 2021-07-02 | 120.71 | 20 | 3.26 | 1.18 | 185 | 4 |
| NoatakSlump | WA | 2021-07-02 | 5.21 | 10 | 5.19 | 2.07 | 34 | 1 |
| NoatakRiverS | WA | 2021-07-02 | 12.94 | 10 | 7.30 | 2.75 | 78 | 1 |
| NoatakCoast | WA | 2021-07-03 | 27.46 | 10 | 10.21 | 3.58 | 138 | 1 |
| Kivalina | WA | 2021-07-03 | 4.14 | 10 | 3.95 | 1.46 | 23 | 1 |
| SPNorthDTLBEast | WA | 2021-07-09 | 22.13 | 10 | 8.95 | 3.16 | 117 | 1 |
| SPNorthDTLBWest | WA | 2021-07-09 | 33.69 | 10 | 8.88 | 3.11 | 179 | 2 |
| SPNorthKitlukCoast | WA | 2021-07-09 | 97.67 | 10 | 9.38 | 3.17 | 460 | 4 |
| SPCKougarok01 | WA | 2021-07-10 | 109.45 | 10 | 8.18 | 2.88 | 549 | 5 |
| KobukDelta | WA | 2021-07-10 | 84.14 | 20 | 2.57 | 1.19 | 108 | 2 |
| SPCImuruk | WA | 2021-07-10 | 84.72 | 10 | 11.21 | 4.00 | 434 | 5 |

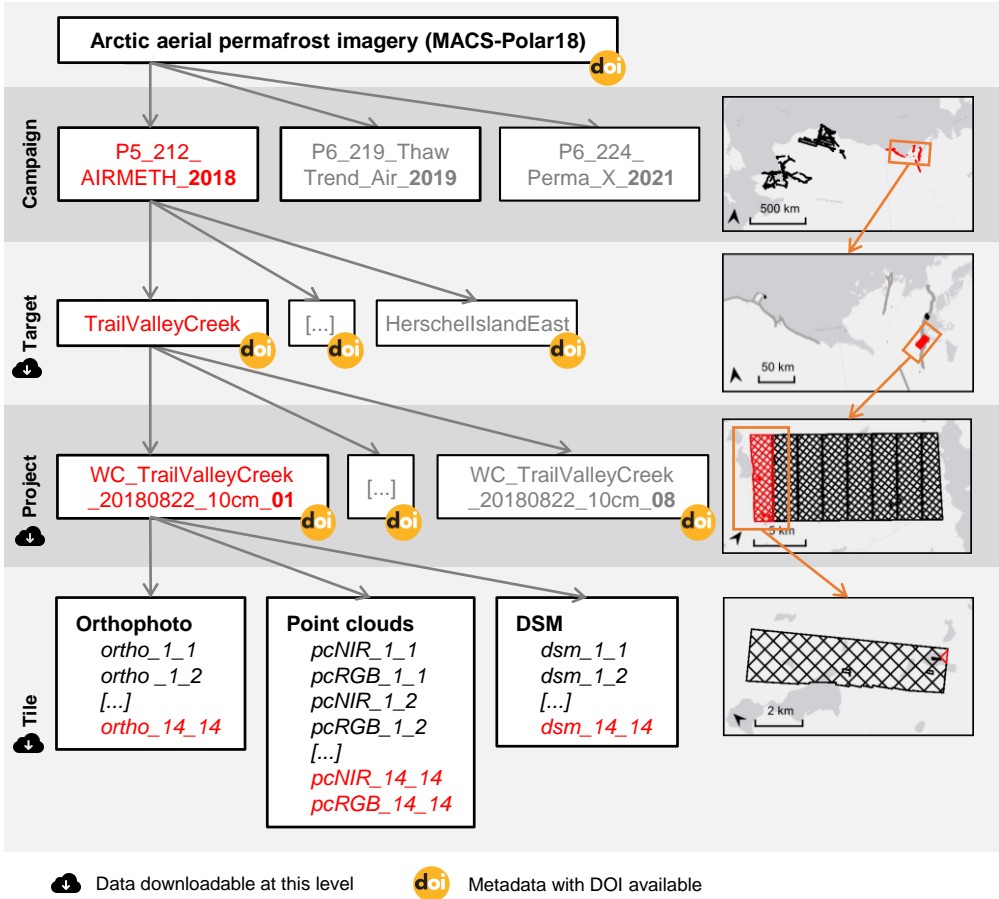

**Figure 6.** Hierarchical structure of the published datasets. This publication includes data from three aerial campaigns. Each campaign covered multiple targets of interest. Due to processing/computational limitations, we split large target areas into multiple processing projects. For easier data handling, orthomosaics, point clouds, and DSMs from each project are tiled into tiles of 5000 px by 5000 px each

(Tuktoyaktuk, Northwest Canada). The mosaics were processed based on the geopositioning information recorded by the GNSS receiver and the IMU of the camera system itself. Their realtime positional errors are in the range of 1 m vertically and 0.6 m horizontally, with 0.02° uncertainty for roll and pitch and 0.1° for heading. For the resulting datasets, this translates to ca. 0.8 m positional error in the horizontal plane and up to 2 m along the z-axis. In a subsequent version of reprocessing these
datasets, we will post-process the GNSS and IMU information of each individual image based on the aircraft's more precise IMU geolocation system prior to calculating the data products. Figures 7, 8, and 9 show example orthomosaics and DSMs of the published datasets at the three different target resolutions.

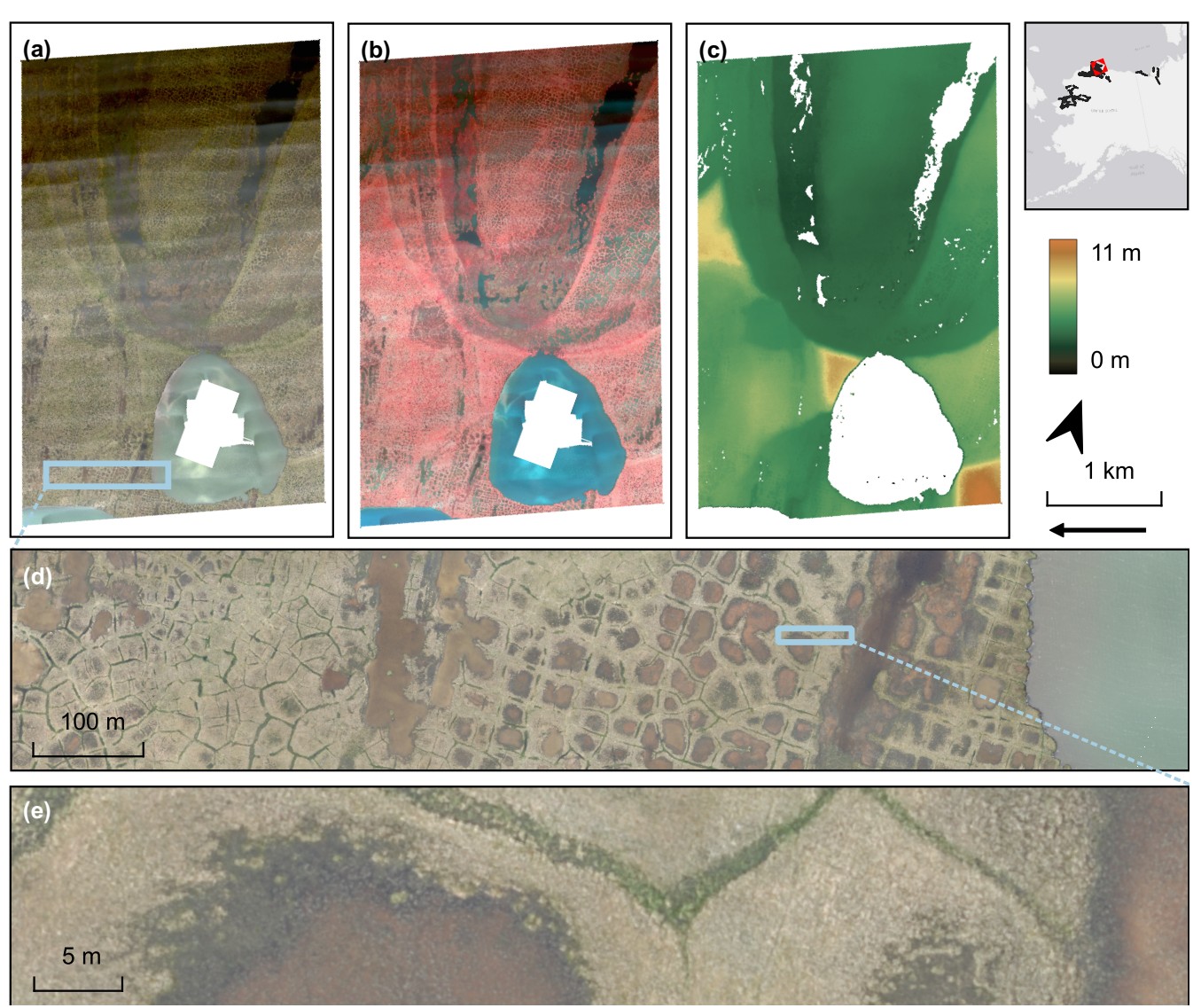

**Figure 7.** AOI *TeshekpukLakeNorth* subset *09* captured on 23 July 2019 and processed to datasets with 7 cm spatial resolution displayed as (a) optical ortho-image in the bands blue, green, and red; (b) color-infrared image in the bands green, red, and near-infrared; and (c) DSM. (d) and (e) show zoomed in details of the optical ortho-image to showcase the level of detail visible in these high-resolution datasets. Basemap in overview panel: Esri, HERE, Garmin, © OpenStreetMap contributors, and the GIS user community.



**Figure 8.** AOI *NoatakSlump* captured on 2 July 2021 and processed to datasets with 10 cm spatial resolution displayed as (a) optical ortho-image in the bands blue, green, and red; (b) color-infrared image in the bands green, red, and near-infrared; and (c) DSM. (d) and (e) show zoomed in details of the optical ortho-image to showcase the level of detail visible in these high-resolution datasets. Basemap in overview panel: Esri, HERE, Garmin, © OpenStreetMap contributors, and the GIS user community.

**Figure 9.** AOI *BPSouth* subset *03* captured on 28 June 2021 and processed to datasets with 20 cm spatial resolution displayed as (a) optical ortho-image in the bands blue, green, and red; (b) color-infrared image in the bands green, red, and near-infrared; and (c) DSM. (d) and (e) show zoomed in details of the optical ortho-image to showcase the level of detail visible in these high-resolution datasets. Basemap in overview panel: Esri, HERE, Garmin, © OpenStreetMap contributors, and the GIS user community.

## 4.2 Photogrammetric point clouds

The processed point clouds are provided as las-files, both in the RGB and NIR bands (Rettelbach et al., 2023). They are
provided as tiles with the same tiling scheme as the orthophotos and also published in UTM CRS, allowing for efficient organization and management of the datasets. Each point in the point cloud contains multiple attributes, including RGB or





NIR reflectance information, as well as the X, Y, and Z location in meters. The density of points within the point cloud varies depending on the spatial resolution of the source images used for point cloud generation and ranges between 2.57 and 48.03 px/m$^3$ for the RGB and 1.18 and 17.85 px/m$^3$ for the NIR clouds.

### 4.3 Additional data

For quick insights and simpler, on-the-fly processing, we also provide pre-computed overview files (pyramid layers) for each individual tile at the respective spatial resolutions, again in four bands. For each data product (orthophoto and DSM), we provide mosaic files in Virtual Raster Tile format (.vrt) which load all tiles at once. We further make available three very-low-resolution, PNG-formatted preview images of the target area. One corresponds to the orthophoto rendered in the optical R-G-B wavelengths, another is a color-infrared preview with the NIR-R-G bands. The third is based on the DSM. These files are suitable for quick-look purposes only and they should not be used for scientific analysis. Some quick-look preview-PNGs have vertical striping as an artifact. This is however only an artifact in the PNG file, and is not found in the original orthophotos or DSMs. All additional files are also available via Rettelbach et al. (2023).

## 5 Data quality assessment

The quality parameters of the output datasets are described in the Pix4Dmapper quality reports generated for each dataset. Additionally, we conducted a visual inspection of the resulting data. The automatically generated Pix4Dmapper quality reports contain information on the technical processing details and the quality of the orthomosaicking process, such as the number of calibrated and matched images, the number of tie points found and considered, the internal camera parameter uncertainties, and the geolocation variance of the individual input images. The quality report is provided with each target dataset and also contains information on the computing infrastructure we used for the processing, the coordinate reference system, as well as the spatial resolution of the resulting orthomosaics, elevation models and point clouds. General guidelines for interpreting the report are provided as an online resource by Pix4D: https://support.pix4d.com/hc/en-us/articles/202558689-Quality-Report-Help. The subsequent visual assessment of the output datasets was conducted to assess the visual quality of the datasets. While processing parameters can be tightly controlled, the image quality is also determined by the lighting conditions during image acquisition, which varied across the targets and in the case of large targets that were acquired over several flight hours also within the grids. Below, the most prevalent issues are described in detail.

### 5.1 Multispectral image matching

The MACS records spectral information in the RGB and NIR wavelengths with separate sensors of different resolutions and FOVs, and thus, different footprints on the ground (see Sec. 2.2 and Fig. 2b). When photogrammetrically processing the images to orthomosaics, Pix4Dmapper matches brightness levels between images of a sensor (RGB or NIR), but is not able to match colors between the two cameras. This can lead to problems when calculating spectral indices that require input from the NIR and an optical band, such as the normalized difference vegetation index (NDVI), requiring information from the NIR and red

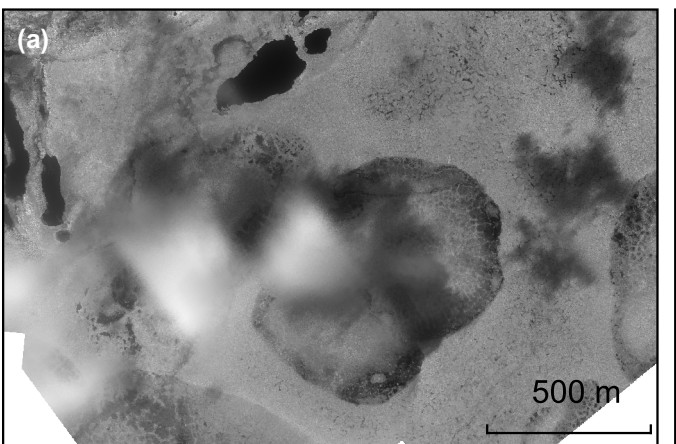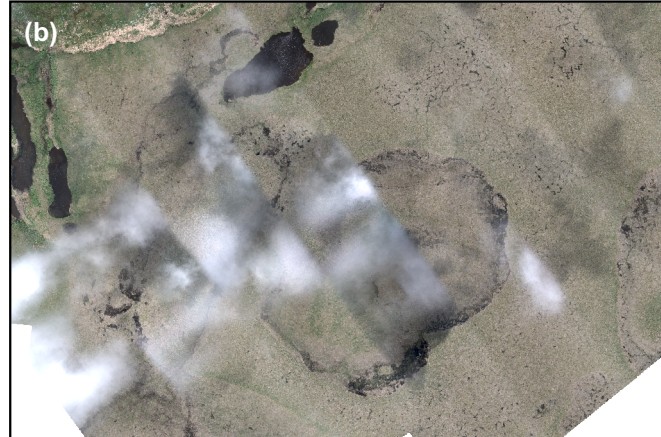

**Figure 10.** Cloudy scenes in MACS orthomosaics may appear different in the (a) NIR and the (b) RGB orthomosaic. While NIR and RGB cameras always captured data simultaneously and thus produced images of the same conditions on ground, the RGB and NIR orthomosaics were computed independently from each other, which could lead to different images instances from different flight lines being incorporated into the RGB versus the NIR mosaic. In addition, if objects such as clouds, their shadows, or waves on water within the imaging footprint moved between the acquisition flightlines, photogrammetric matching usually failed and flight lines became very apparent through cut-off objects such as the clouds in (b). Image location: Northern Seward Peninsula at 66.5370° N, 164.0700° W. Image taken on 2021-07-09.

wavelengths. The effect was particularly problematic in inhomogeneously illuminated areas, such as at the overlap of two flightlines when the illumination changed between the two tracks (from e.g., shifting clouds in the meantime). In such cases,

the mosaicking algorithm might favor images from different flightlines for the same pixel in the NIR and in the RGB. Fig. 10 illustrates how, for example, clouds appear differently in RGB and NIR orthomosaics as they have moved between the timing of the flightlines.

### 5.2 Changing illumination

Prior to any acquisition flight, we set the camera parameters according to the prevailing illumination conditions. Heavy cloud

cover with homogeneous diffuse light, for example, required longer sensor exposure times than clear skies and direct sunlight. Despite our best efforts, no single camera exposure setting could compensate for the slight changes in illumination between flight lines of a large target area. As post-processing cannot entirely mitigate this effect either, such brightness differences also manifested in the orthomosaics along the edges of flight lines. Fig. 11a shows the optical orthomosaic of a tundra landscape near the village of Buckland with images collected on 27 June 2021 and gives an example of this data artifact. In most ortho-

images these flight lines are to some degree visible; however, especially on a local scale, they become almost negligible. The derived DSMs are only affected in very rare cases, where illumination changes between flight lines were extreme. This can be seen in the dataset of the Ketik fire scar, as displayed in Fig. 11c and d.

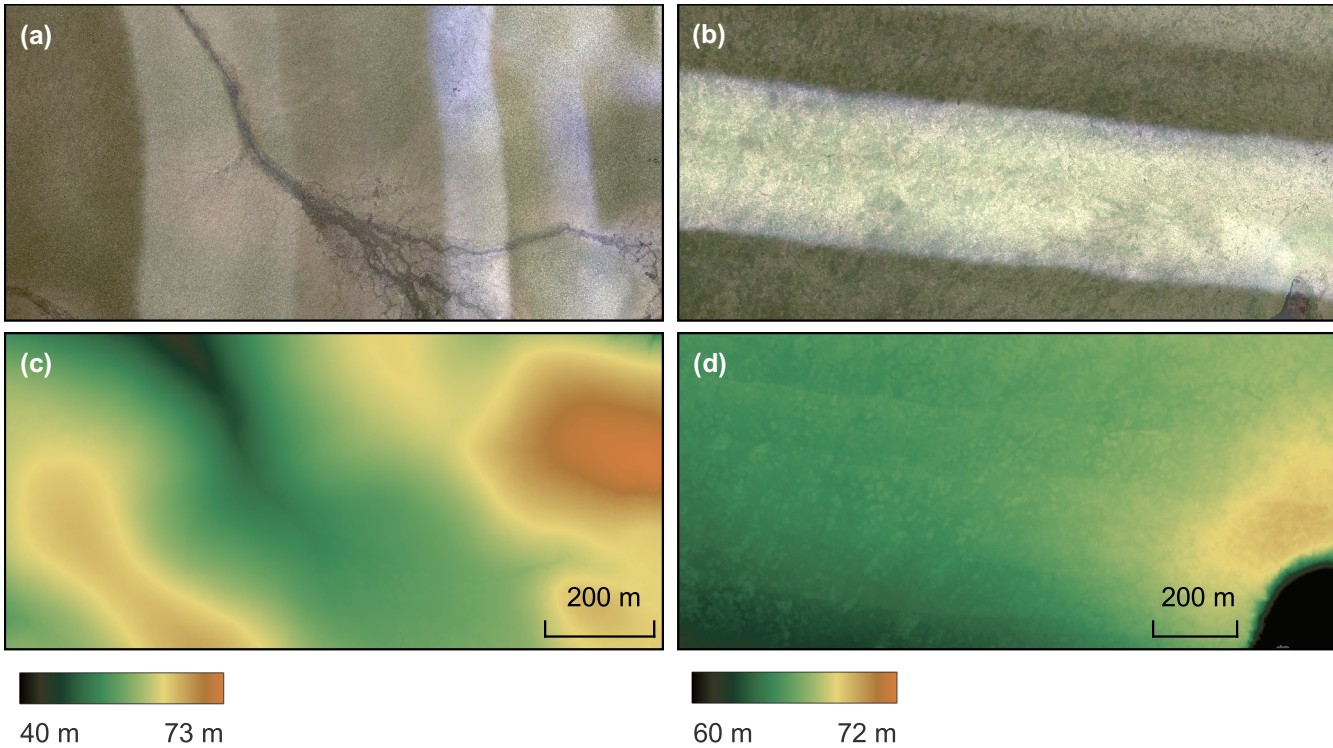

**Figure 11.** (a) and (c) show RGB orthomosaics generated from images with strong illumination differences between neighboring flight lines. (b) and (d) show the corresponding digital surface models (DSMs) of (a) and (c) respectively. Even though orthomosaics often show these striping artifacts caused by illumination differences, they only manifest in DSMs in cases of very extreme differences in image brightness. Image location (a) and (c): Buckland fire scar at 65.9697° N, 161.0475° W. Image taken on 2021-06-27. Image location (b) and (d): Ketik fire scar at 69.9149° N, 159.3557° W. Image taken on 2019-07-27.

In extreme cases, such as during some acquisition flights in Northern Alaska in 2019, some images had very strong overexposure. Especially in the NIR images, we then find a 'smear' effect around the centers of the affected acquisitions (see. Fig. 12a). This is an artifact resulting from the sensor design: When too many photons reach the interline charge-coupled device (CCD) sensors, the buffer overflows into the next line, creating what looks like vertical streaks in the resulting image. Accordingly, the saturation for these pixels is at 100 %. It is thus also not possible to post-process the affected images in any way to regain the spectral information of the underlying landscape. For flights during the later campaign in West Alaska in 2021, we captured imagery using multi-exposure high-dynamic range (HDR, usually at 1 ms and 0.4 ms) whenever inconsistent cloud cover seemed probable. With this setting, we were able to decide later which of the two exposure times generated the better image for each individual target area and only considered those for further processing.

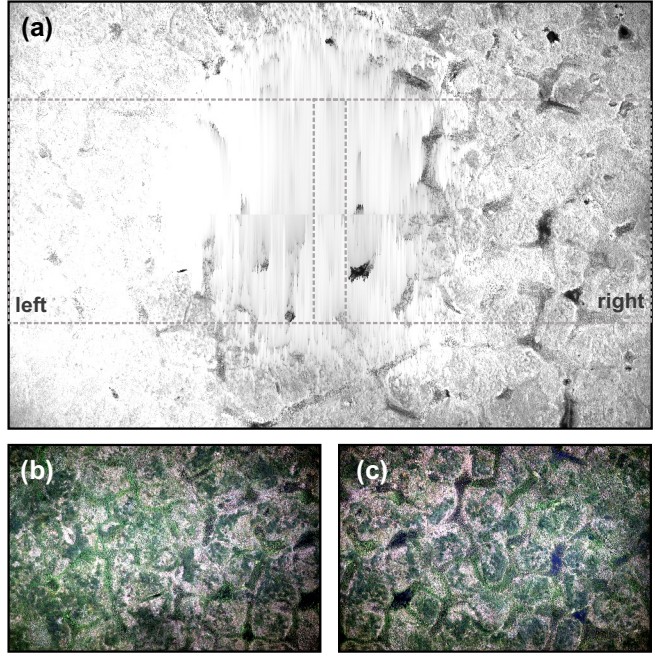

**Figure 12.** (a) Original MACS NIR image shows a smear effect at the center of the image. Due to overexposure, the affected pixels are entirely saturated and the CCD sensor buffers overflow into the next line, oversaturating the next pixel too. The RGB images of the (b) left- and (c) right-looking cameras were set to more adequate exposure times and do not show any smearing. The dotted rectangles in (a) show the left and right RGB footprints. Image location: Polygonal tundra in Ketik River fire scar at 69.9107 ° N, 159.2578 ° W. Image taken on 2019-07-27.

## 5.3 Flight patterns

To obtain high quality 3D point clouds and DSMs, it is generally recommended to use an overlap of minimum 60 % across- and 75 % along-track during flights (Pix4D, 2021b). The flights over TVC in Canada in 2018 were however flown at only 20 % across-track overlap, which has impacted the quality of the processed DSM. While all other targets where flown in a regular grid pattern, where neighboring flightlines where acquired directly one after the other, with alternating flight directions, the TVC target was flown in loops. Figure 13e shows the order and flight direction of the lines for the aerial grid. For this extreme case, we conducted a comparison of one of our TVC DSM subsets (Fig. 13b) with the airborne laser scanning (ALS) digital terrain model (DTM, Fig. 13c) that was acquired during the same flight (**?**). The ALS DTM is of high quality with an accuracy of 0.03 m and a precision of 0.08 m. We applied the DEMcoreg algorithm (Shean et al., 2016) which is based on the method outlined in Nuth and Kääb (2011) to first align the MACS DSM to the ALS DTM and then conduct differencing. We found a vertical offset of -1.40 m and a horizontal offset of -0.03 m in x-direction and -3.44 m in y-direction (see Fig. 13). The strongest elevation differences can be found in the center of the target, where flightlines 2 and 17 meet. They where captured ca. three hours apart from each other. We attribute these strong mismatches to the shift in illumination between the acquisition times





**Figure 13.** AOI *TrailValleyCreek* subset *03* with data from 22 August 2018. (a) Optical orthophoto; (b) photogrammetric DSM; (c) DTM generated from ALS data. (d) shows the elevation difference between the MACS- and ALS-derived elevation models with a strong height mismatch in the center. (e) has overlain the flight pattern (order and directions of the flightlines) to emphasize the impact that the timing of adjacent flightlines can have on the photogrammetric processing. Image location: 68.6994 ° N, 133.6874 ° W.

and to the opposing viewing directions of the sensors, where they meet. With the very low across-track overlap and the looping flight pattern, this mismatch could not be corrected from further outward-lying, neighboring flightlines (i.e., 4 and 15).

## 5.4 Water areas

Typically, water areas cannot be matched through tie points in SfM, as the contrast in the imagery is too low. Resulting point clouds therefore usually have no or extremely sparse data for these areas. During the calculation of DSMs from point clouds (see Sec. 3.3), we did fill small holes by interpolating them, but most water areas exceed this size and therefore we get large no-data areas. This results in no-data gaps within the DSMs where areas were covered by larger water surfaces. Fig. 14 shows



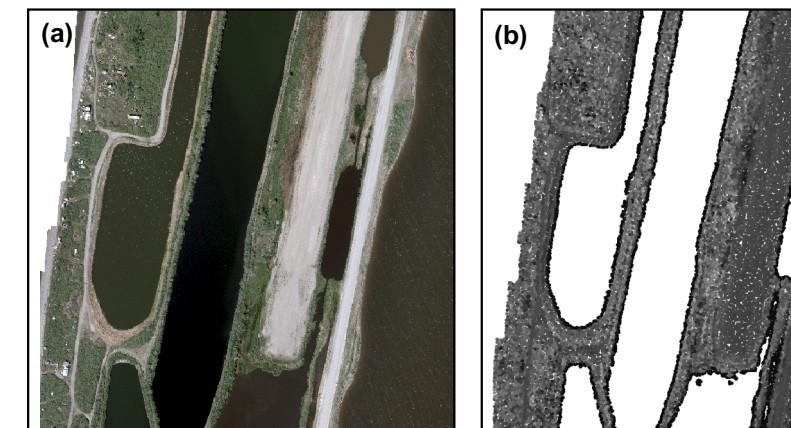
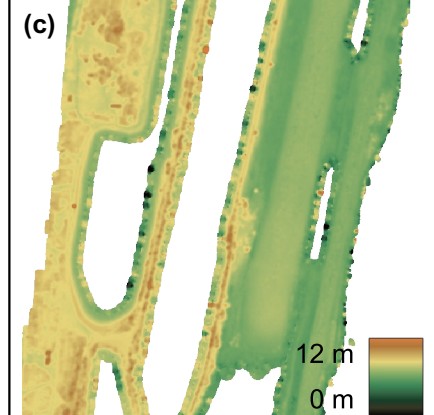

**Figure 14.** (a) Multispectral orthophoto of an area near Kotzebue characterized by water areas. (b) Point cloud of the same area showing the large data gaps of the water areas from the ocean, ponds, and lakes. (c) In the resulting DSM, the water areas therefore also represent areas of missing data. Image location: Kotzebue spit south of the Ralph Wien Memorial Airport on Baldwin Peninsula, West Alaska at 66.8756 ° N, 162.6167 ° W. Data collected on 25 June 2021.

an example with DSM and point cloud data gaps from the Kotzebue spit south of the airport. This area is characterized by ponds and lakes on the narrow spit between the Chukchi Sea in the West and the Kotzebue Lagoon in the East.

In addition, for waves and whitecaps in the ocean or in wind-blown lakes and rivers, the color contrast does allow matching during the SfM processing. Therefore, these areas generate sufficient points within the dense point cloud to interpolate when creating the DSMs with WhiteboxTools. We see this effect for example in the dataset of Shishmaref, a village on an island in the Chukchi Sea (Fig. 15). It is important to note that the DSM of such water surfaces is however not representative of different wind and wave conditions and these areas should thus not be used for analysis. In a future release of the dataset, we aim to implement a robust algorithm to mask out these water areas.

## 6 Conclusions

We here publish and describe super-high-resolution aerial datasets of permafrost landscapes covering more than 6000 km$^2$ in northwestern North America (Rettelbach et al., 2023). These datasets were derived from the optical and near-infrared MACS sensors during three airborne campaigns with the AWI airplanes Polar-5 and Polar-6. They include photogrammetric orthomosaics, point clouds, and digital surface models at spatial resolutions from 7 to 20 cm GSD or 1.18 to 48.03 px/m$^3$ point cloud density. These datasets provide an extraordinary level of spatial resolution for a wide range of observation targets in permafrost regions of North and West Alaska and Northwest Canada. The high spatial detail opens up new possibilities for data analysis, as well as discovery, and validation of small-scale permafrost landforms such as ice-wedge polygons and their subtle morphology and typical northern land surface features such as individual shrubs and trees or beaver lodges. Apart from providing the base for mapping specific features, their distribution, and their microtopography at target sites, the datasets will be useful

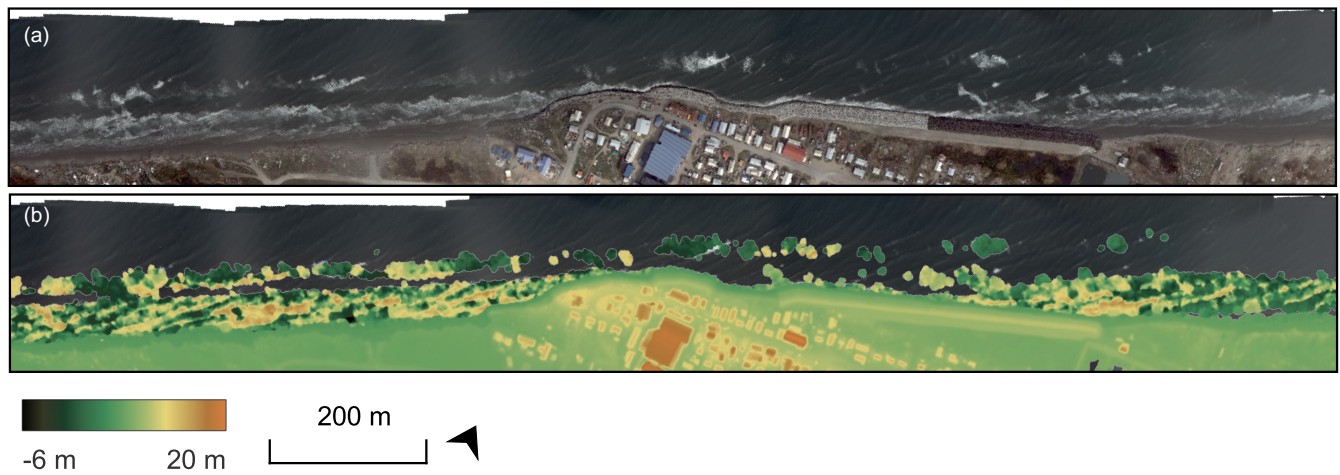

**Figure 15.** (a) shows the optical orthophoto of the Chukchi Sea coast in the village of Shishmaref (Qigiqtaq) with waves and whitewater along the entire coast. (b) overlays the digital surface model (DSM) of this area, showing the response of these disturbed waters in the point cloud and thus in the resulting DSM. Image location: Village of Shishmaref (Qigiqtaq) on Seward Peninsula, West Alaska at 66.2557 ° N, 166.0731 ° W. Data collected on 28 June 2021.

to create super-high-resolution training datasets for machine-learning algorithms in support for lower-resolution satellite imagery. With this, the MACS-derived datasets build a highly valuable foundation for scaling analyses and for change detection with historic and future airborne or satellite datasets. For several communities in West Alaska, the datasets provide recent super-high-resolution baseline imagery for assessing infrastructure risks from thawing permafrost nearby or for community infrastructure planning efforts enhancing adaptation and allowing some mitigation of consequences of climate change impacts.

For the upcoming years 2023-2026, we plan to expand this dataset spatially and temporally through revisits in these three regions during upcoming aerial campaigns, allowing to cover existing sites again for change detection analysis and to also add new sites of high interest. Overall, these super-high-resolution datasets can become an essential tool for understanding the impacts of climate change on permafrost regions and can provide insights into the processes and dynamics of rapidly changing permafrost landscapes in the Arctic.

**7   Code and data availability**

All data resulting from the processing described in this manuscript is available to the public via PANGAEA, an Open Access data publisher archiving and distributing georeferenced data from Earth and environmental research. The following DOI represents access to the collection of all datasets covered by this publication: https://doi.pangaea.de/10.1594/PANGAEA.961577 (Rettelbach et al., 2023). DOIs for all 102 individual targets are also available (reachable via this overarching DOI) but for

reasons of readability shall not all be listed in this printed publication.

Raw, individual images from additional flight tracks (see black areas in Fig. 1), not covered by the processed data described in this publication, is available upon demand to the authors until further processing and public data archival may be conducted in the future. During all three campaigns described in this publication, additional sensors i.e., a full-waveform LiDAR (Riegl LMS-Q680i), a slewable camera, an infrared thermometer, as well as sensors for measuring air temperature, moisture, and pressure were installed on the planes and recorded measurements at the same time as the MACS recorded images. For the flights in North Alaska in 2019, a further sensor recorded methane concentration. Data of these additional sensors is available from the authors upon reasonable request.

Some point clouds derived from the LiDAR system of the flights in Northwest Canada in 2018 have already been processed and published on PANGAEA: TrailValleyCreek_20190822 (Lange et al., 2021a) and TukRoadGrid_20190822 (Lange et al., 2021b). All code used to process the here published datasets is available to the public through Github: https://github.com/awi-response/macs_processing. *A long-term Zenodo DOI will follow after review.*

## Appendix A: Point cloud source sensor for digital surface models

As the options to process DSMs from multispectral point clouds are manifold, prior to the bulk processing of all targets, we conducted a parameter search to determine the ideal settings. These tests were made on three datasets that cover different landscape types and permafrost features represented over the entire available MACS image space. The first AOI is found around the Selawik Thaw Slump in West Alaska. This thaw slump shows steep edges, as well as individually standing trees. The second AOI is Kivalina on Seward Peninsula, West Alaska, representing a village with buildings of different sizes. Finally, Teshekpuk Lake serves as the third AOI for comparison purposes. Here we find small elevation differences between the ice-wedge polygons and their troughs in between. It is important that any DSM-generating algorithm can both preserve these fine elevation details, and correctly represent steep or sharp edges, such as from the buildings, the thaw slump's head wall, or the individual trees.

We investigated what sensors generate the best DSM results. Our analyses show that there is no general tendency towards either the NIR or the RGB sensor, but that the matching algorithm performs badly for oversaturated or undersaturated and very dark pixels in the original image. Within our three comparison AOIs, this effect can be seen in the DSMs from the RGB-only point cloud for Kivalina and the polygonal tundra near Teshekpuk Lake. In Kivalina, many metal roofs of buildings show oversaturation in the images and thus complicate the correct matching of pixels. This results in frayed and imprecise building edges. Similarly, the undersaturated water areas from the thermokarst ponds in the polygonal tundra AOI also show imprecise matching in the RGB-only DSM (Fig. A1i). As oversaturation can also be a problem in some NIR images (see Fig. 12), we found targets where the NIR-only point cloud is also affected by this issue. Furthermore, using the NIR-only point cloud, we also observed that the resulting DSMs showed less sharp edges in comparison to the DSMs from the RGB-only point clouds. This is a result of the lower point density of the NIR point clouds. This effect can be seen both at the thaw slumps head wall edge in Figs. A1d and g and the edges of buildings in Figs. A1e and h. Using the combination point cloud (Fig. A1k-m) can overcome the worst of both the described effects and results in the most coherent DSMs for most targets. Only in





such areas, where one of the two sensors presented significant issues, we resorted to generating the published DSMs from the

495 higher-quality single point cloud.

*Author contributions.* Conceptualization: GG, TS, IN, TR; Data curation: IN, TR, SS; Formal analysis: IN, TR, SS; Funding acquisition: GG, TS; Investigation: GG, IN, TR, JB, IG, TS, JH, MG; Methodology: IN, TR, IG, JH, SS; Project administration: GG, TS; Resources: GG; Software: IN, TR, IG, JH, SS; Supervision: GG, TS, JB; Validation: IN, TR, IG, JH; Visualization: TR, IG, JH; Writing - original draft preparation: TR; Writing - review & editing: TR, IN, GG, TS, IG, JH, DH, MG, TB, JB, JB.

*Competing interests.* The authors declare that they have no conflict of interest.

*Acknowledgements.* We acknowledge support for facilitating Polar-5 and Polar-6 airplane logistics and technical preparation by AWI Logistics and Scientific Platforms personnel Daniel Steinhage, Benjamin Harting, Christoph Petersen, Martin Gehrmann, Maximilian Stöhr, Cristina Sans Coll, Eduard Gebhard, Silke Henkel, and Uwe Nixdorf. We thank Birgit Heim for helping with preparation of initial MACS logistics and Katrin Kohnert for campaign and flight planning in 2018. We thank the Polar-5 and -6 captains, first officers, and engineers from

505 Kenn Borek Air for their great service: Dean Emberley, Jamie Chisholm, William Houghton, Kodi Bacon, Matthew Patz, Linden Hoover, Erik Prager, Jamie Harrison, and Ryan Schrader. We are thankful for many US and Canadian partners supporting permitting and survey target selection. We would further like to thank Amelie Driemel and Maximilian Betz from the PANGAEA team for their invaluable support with data and metadata management. The airborne campaigns were facilitated with AWI base funds and additional support was received through ESA CCI+ Permafrost, ESA EO4PAC, EU Nunataryuk, EU Arctic Passion, HGF AI-CORE, HGF HIP, HEIBRiDS and Geo.X.





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





**Figure A1.** Three AOIs representing different landscapes of the dataset domain were chosen to conduct a comparison of DSMs originating from three different data sources. Orthophotos of (a) the Selawik Thaw Slump and (b) Kivalina in Western Alaska, and (c) a polygonal thermokarst landscape near Teshekpuk Lake in Northern Alaska show the three test sites. Below are the resulting DSMs (d-f) from the NIR-only point cloud, (g-i) from the RGB-only point cloud, and (k-m) from the point cloud with both RGB and NIR points.