# Peer review of "Super-high-resolution aerial imagery datasets of permafrost landscapes in Alaska and northwestern Canada"

_Earth System Science Data, 2023_

## Community Comment (CC2)

[supplement omitted: unrelated document]

---

## Author Comment (AC2)

We would again like to thank the anonymous referees and Matt Nolan for their time in reviewing our manuscript. Below, we address each comment with the following formatting for improved readability:

**Referee comments**
Author's response
*Changes made*

Anonymous Referee #1:

**The article is well-written, I have no comments except for one on Figure 3, which is missing the "ng" in the word "processi".**

Thank you for noticing, we have of course adapted this in the updated manuscript!

*Page 11, Fig. 3.*

[Figure]

Other than that, the authors might consider citing four articles that are valuable from the point of view of thermokarst lakes:

Chen, X., Mu, C., Jia, L., Li, Z., Fan, C., Mu, M., Peng, X., & Wu, X. (2021). High-resolution dataset of thermokarst lakes on the Qinghai-Tibetan Plateau. Earth System Science Data Discussions, 1–23.

Hughes-Allen, L., Bouchard, F., Laurion, I., Séjourné, A., Marlin, C., Hatté, C., Costard, F., Fedorov, A., & Desyatkin, A. (2021). Seasonal patterns in greenhouse gas emissions from thermokarst lakes in Central Yakutia (Eastern Siberia). Limnology and Oceanography, 66(S1), S98–116. https://doi.org/10.1002/lno.11665.

Janiec, P., Nowosad, J., & Zwoliński, Zb. (2023). A machine learning method for Arctic lakes detection in the permafrost areas of Siberia, European Journal of Remote Sensing, 56:1, 2163923, DOI: 10.1080/22797254.2022.2163923.

Wu, Y., Duguay, C. R., & Xu, L. (2021). Assessment of machine learning classifiers for global lake ice cover map ping from MODIS TOA reflectance data. Remote Sensing of Environment, 253, 112206. https://doi.org/10.1016/j.rse.2020.112206.

The focus of our manuscript lies on the provision and description of aerial image datasets that cover large areas in the permafrost domain. While these datasets include many areas that are abundant in thermokarst lakes, we do not set a stronger focus of the manuscript on this permafrost feature compared to any others. We agree with the referee that there are many suitable papers to reference on the topic of thermokarst lakes, including the ones suggested. However, we have already provided three exemplary recent studies (Nitze et al. 2020, Lara et al. 2021, Jones et al. 2020) that we believe cover a broad range of the topic. We hope that the referee and the editors can agree with our exemplary choice of cited studies as well.

Anonymous Referee #2:

**General comments:**

**This paper describes super-high-resolution aerial imagery datasets of permafrost landscapes in Alaska and northwestern Canada. To the best of our knowledge, acquiring aerial remote sensing imagery involves a substantial investment of human and financial resources. Consequently, the diverse datasets provided by this study offer robust support for a multitude of research endeavors. The authors have comprehensively expounded on various aspects, including flight design, data preprocessing, product generation, and product release, effectively showcasing intricate procedural details to the readers. The paper exhibits a well-structured format, clear logic, and authentic English expression, rendering it a high-quality scientific contribution. Nonetheless, a few queries and suggestions persist, and I would greatly appreciate it if the authors could address them.**

Thank you very much, we greatly appreciate the time the anonymous referee has put into the review of our manuscript. We hope that in the following, we can address all concerns in a satisfactory manner.

**Specific comments:**

**1. In the Abstract, the authors describe parameters such as spatial resolution and point cloud density of the generated datasets. However, there is no mention of an overview of the dataset size and specific product accuracy. It is recommended that the authors include a brief description of the product quantity (e.g., the total number of orthophotos and the number of point cloud datasets) as well as the product quality (e.g., geometric errors, visual quality of the images, etc.) to provide readers with a more intuitive presentation.**

Thank you for pointing this out. We agree that this information should already be mentioned in the abstract. We have extended it accordingly. We have also included a column in the newly added Table A2 that shows the number of single images that went into creating the orthomosaics to get an idea of the dataset quantities.

Added on page 1, line 12: *"Project sizes range from 4.8 GB to 336 GB. In total, 3.17 TB were published. Geometric accuracies of the datasets are in the range of 0.28 m ± 0.12 m (horizontal precision) and 0.18 m ± 0.06 m (vertical precision). The datasets are not radiometrically calibrated. As such, these very-high-resolution images and point clouds provide significant opportunities for [...]."*

Added on page 36, Table A1, column "*# raw images*"

**2. Page 4, Figure 1. The black lines in the graph appear to be somewhat irregular and contain breakpoints. Could the authors explain the significance of designing flight paths in this manner? Additionally, what are the factors that lead to interruptions in the flight route?**

The black lines in the map show the *footprints of all acquired images* with the MACS during the respective campaigns. As such, they do not automatically represent the entirety of flight *paths* that have been flown. The "breakpoints" therefore represent areas that we did fly, but where the multispectral camera was not operating due to changes in the setup and thus not acquiring images. We have added two sentences in the section on survey design to clarify this. We have also updated Fig. 1 in accordance with the community comment (see also CC #20) to set a stronger focus on the here published targets (now in pink), rather than the entirety of acquired data.

Added on page 9, line 213: *"We acquired images for all transit flights to, from, and in between planned target grids. The camera was only turned off during take-off and landing, when low-level clouds occurred locally, over larger water bodies or sensitive infrastructure, during sharp turns, or when space on the hard drive for data storage was running low."*

Updated Fig. 1, page 4:

[Figure]

**3. Page 6, lines 146-148. The authors mention that rainfall may affect the state of water bodies and the local hydrological conditions. Did the authors take into consideration the characteristics of rainfall when designing the flight paths?**

Flight paths for these large-scale aerial campaigns were planned ahead of the surveys, based on the permafrost characteristics and features in a certain area. As such, the (temporally) local rainfall characteristics have not been taken into consideration for the detailed planning of the flight paths. In the manuscript we considered it worth noting that the exceptional rainfall in July 2021 contributed to certain datasets depicting atypical hydrological conditions instead of those from an average year. However, it is important to highlight that imaging flights exclusively occurred under precipitation-free conditions. We have added this information to the manuscript:

Add on page 6, line 151: *"However, in 2021, the year we surveyed, the area experienced record-breaking 137.9 mm of rainfall in July alone (NOAA, 2023). This potentially altered the average-year water levels of water bodies and the overall hydrological state depicted in the acquired datasets of that year. Nevertheless, imaging flights were exclusively conducted during precipitation-free conditions."*

**4. Page 6, line 152. The authors mention that the MACS sensor is specifically designed for the tough environment of the Arctic region. What distinguishes this device from typical equipment? While the author has provided references, it is recommended to briefly describe in the main text the reasons for the suitability of this device for the Arctic region.**

The key points that allow the MACS to operate in the Polar regions are its suitability to function in low temperatures and the systems capability to capture at multiple different shutter speeds. We generally revised the Section **2.2 Multispectral sensor** and have highlighted these points in the updated manuscript. See also CC #12.

Add on page 7, line 157: "For all three campaigns, we used the custom-built Modular Aerial Camera System (configuration: MACS-Polar18) developed by the Institute of Optical Sensor Systems of the DLR. *It was specifically adapted to work in very low ambient temperatures. Multiple integration times per scene can be acquired to avoid under- or over-exposed pixels in the challenging Arctic light conditions, where very dark (water, dark bare soil) and bright (snow, ice) surfaces often co-exist in target areas (Brauchle et al., 2015).* [...]"

**5. Page 8, lines 182-184. The authors have only provided grid-stitched data and have not presented strip-stitched data. Based on my experience, stitching strip data from UAV or manned-aircraft flights can be more challenging than grid data, and it often results in significant missing when using automated stitching software like Pix4D. Did the authors encounter this issue during data processing? If so, have you undertaken any specific measures to address it?**

We agree with the referee that stitching or mosaicking strip-flown data is quite a bit more challenging than mosaicking grid-flown data. Strip-flown data only provides along-track overlap between images and lacks images in adjacent swaths that would provide additional viewpoints and across-track overlap. It is for this reason that we only provide the stitched grid-flown data in this first version of the processing and dataset publication. We have made this distinction clearer, including in Figure 1.

Added on page 10, line 216: "While we here only publish datasets generated from targets that we flew in grid-patterns which were suitable for photogrammetric processing (labeled in yellow in Fig. 1a-c), we also did capture additional imagery on single-track transect flights (labeled in black in Fig. 1; master tracks available via Hartmann (2018) and Grosse et al. (2019, 2021)). *[...] Tests of processing strip-flown data resulted in mosaics with stronger bowling effects and distortions, especially at the borders of the images, that could not automatically be corrected due to lack of images from neighboring tracks. However, it is possible to manually correct for such effects in the post-processing stage. For the large volumes of our collected data, this manual correction for all datasets was however not feasible at this stage. Nevertheless,* the raw, individual images from additional flight tracks (see black areas in Fig. 1), not covered by the processed data described in this publication, is available upon demand to the authors until further processing and public data archival has been conducted."

**6. Page 9, line 200. What specific aspects are included in the "cleaning operations"? Were these operations carried out manually or automatically using software or programs?**

With the referenced sentence "[...] we applied pre-processing and cleaning operations and exported [...]. The pre-processing steps are described in the following.", we intend to provide a transition to the following subsections 3.1.1 and 3.1.3, which report in detail the pre-processing and cleaning operations conducted. We have modified it slightly to make it obvious that cleaning operations (such as devignetting) are also described in more detail in the subsubsections 3.1.1 - 3.1.3 of this subsection 3.1:

Added on page 12, line 249: "[...] we applied pre-processing and cleaning operations and exported [...]. *Cleaning and* pre-processing steps are described in the following."

**7. Page 9, line 206. In the flight experiment, RGB and NIR band data were collected. Are the DSNU parameters used consistent for different bands? What determines the choice of these parameters?**

The DSNU parameters for the different sensor CCDs were calibrated in a lab setting before installation into the aircraft. To measure this pixel-dependent sensor noise, we acquired images with mounted lens caps to the cameras in order to avoid any illumination. The given offset, also known as bias, varies with the exposure time and is later applied during MIPPS image preprocessing. We've added a short sentence on this step to Sect. 3.1.1.

Added in page 12, line 254: *"The given offset, also known as bias, varies with the exposure time. The sensors' DSNU parameters were calibrated beforehand in a laboratory setting by acquiring images with mounted lens caps to avoid any external illumination."*

**8. Page 9, line 209. I would like to express my significant concern: The authors have decomposed the original RGB images into three bands. Can each of these bands quantitatively reflect the radiometric information of the Earth's surface, or are these band values relative? If it is the latter case, the application scenarios for the "multispectral" data obtained by the authors will be greatly limited, perhaps only supporting qualitative research rather than quantitative research. In my experience,**

**obtaining accurate surface reflectance information requires the use of ground-based calibration panels, which seems to be lacking in this study.**

Unfortunately, we did not have the opportunity to set out ground-based calibration panels and thus, the datasets are not calibrated for radiometric calculations for e.g., environmental indices as of now. The recorded values are digital numbers and the bands' values are relative to each other. Major calibration and manual data post-processing steps are required to guarantee the radiometric accuracy and suitability for band calculations. For now however, we focus on the datasets' value for many other applications (see below). We propose to include a clear statement about the radiometric limitations both in the Abstract and in the Conclusions, making sure that the readers are in the clear about this.

Added on page 1, line 14: "[...] *The datasets are not radiometrically calibrated. As such, these very-high-resolution images and point clouds provide significant opportunities for [...].*"

Added on page 31, line 539: "[...] beaver lodges. *Given the absence of radiometric calibration targets, calculating radiometric indices may not be wholly reliable with the current version. However, it is important to note that these constraints do not diminish the datasets' utility for object-based analyses, automated segmentation tasks or the mapping of specific features, their distribution, and their microtopography.* The datasets [...]"

**9. Additionally, if possible, please provide the central wavelengths and full-width half-maximum (FWHM) information for the R/G/B/NIR bands.**

The FWHM of the wavelength ranges for all four bands can be found in Table 1, page 9. We have also populated the table with additional specifications in accordance to CC #13.

Updated Table 1, page 9:

**Table 1.** Technical specifications of the MACS-Polar18 configuration.

| Frame rate max. | 4 fps continuous | |
|---|---|---|
| Operating temperature min. | -20 °C | |
| Weight | 17 kg | |
| INS* | GNSS Novatel OEM6 / IMU Sensonor STIM300 | |
| | **2 × RGB sensor** | **1 × NIR sensor** |
| Wavelength range (FWHM†) | B: 450-510 nm
G: 490-580 nm
R: 580-700 nm | 715-950 nm |
| Dynamic range | 62 dB | 62 dB |
| Sensor resolution / pixel pitch | 16 MP / 7.4 $\mu$m | 16 MP / 7.4 $\mu$m |
| Focal length | 90 mm | 50 mm |
| IFOV‡ | 81 $\mu$rad | 140 $\mu$rad |
| FOV§ (cross × along track) | 23° × 16° per sensor
44° x 16° both sensors | 40° × 22° |
| Swath width (at 1000 m AGL) | 400 m | 700 m |
| GSD (at 1000 m AGL) | 8 cm | 15 cm |

* inertial navigation system; † full width at half maximum; ‡ instantaneous field of view; § field of view

**10. Page 10, Figure 3. In the image fusion process, what method was used for blending overlapping areas of images? (e.g., "blending", "averaging", etc.)**

To fuse the different images together, Pix4Dmapper uses a blending process. Unfortunately, Pix4Dmapper, as a proprietary software, does not provide any concrete details on parameters for their algorithms. From a tutorial video, we might be able to assume that the software uses a "blending" approach to fuse together the individual images. However, this source is rather vague. We have reached out to the company to inquire for details on the specifics, but they have repeatedly stated that they do not share details on this. As this is quite an unsatisfactory situation, we aim to slowly shift to processing all datasets with open-source software in the long term.

**11. Page 12, line 239. The authors mention creating multiple subprojects, but was color correction and geometric correction applied to the orthophotos generated from these subprojects to facilitate their subsequent applications by users? In other words, are the images ready for use without any additional processing, or do they require special treatment?**

While the individual subprojects of a target were handled separately, and thus not specifically corrected to match each other, they do line up quite well in terms of geometric and color characteristics. We have ensured that all subprojects have sufficient overlap between each other, so that users that wish to work with multiple subprojects can easily apply merging and matching algorithms most suitable for their purposes.

**12. Page 15, line 334. When the author standardized the spatial resolution of the images, which upscaling algorithm was used for the data with higher spatial resolution? Different upscaling algorithms may be suitable for different image data types.**

The process of homogenizing the slightly different spatial resolutions of the images happens within the Pix4Dmapper processing chain. However, similar to comment #10, this information is unfortunately not provided by the software company. Attempts to contact them and inquire for details were unfortunately not fruitful. We assume however, that the spatial resolution for the output is derived from the point clouds and not from the images directly. Given that we have no reliable source for this, we have not included more information on this into the manuscript. We hope that this is understandable and apologize for the situation.

**13. Page 18, Figure 7. There appear to be horizontal stripes in the stitched image. What is the reason behind these stripes? The spacing between these stripes seems regular and not consistent with the explanation given in section 5.2, "Changing illumination". Is there a method to remove these stripes?**

The regular horizontal stripes mainly stem from the flight pattern and each stripe can be attributed to a flightline. Especially when one flightline is directed towards the sun, and a neighboring flightline is flown in the opposite direction, this effect comes into play. It stems from the differences in the BRDF (bidirectional reflectance distribution function). We have added an additional chapter on this issue.

Changes made line 511, page 29:

*"**5.2.3 Illumination angle and bidirectional reflectance distribution function**

A second source for visible linear artifacts stems from the bidirectional reflectance distribution function (BRDF). The BRDF describes how surfaces reflect light at different angles of incidence and reflection. Therefore, when a flightline is directed towards the sun and the neighboring line is flown in the opposite direction (away from the sun), this variation leads to changes in the perceived reflectance of the surface. Surfaces with different BRDF characteristics will reflect light differently based on the flightline angle. This variation affects the radiometric properties of the captured imagery, causing variations in brightness, contrast, and spectral response across the images (examples can be seen in Figs. 7 and 13a).*

*Processing software will often select high-contrast features to prepare for image matching, which may include shadows cast on the ground. So, the more time that has passed between two flightlines that should be matched photogrammetrically, the larger the induced error, as any shadows wander across the ground with a changing sun illumination angle.*

*The safest way to avoid such artifacts is by already factoring in the sun position into the flight planning phase. In our case, this was often not possible: Preparing flight plans for airborne surveys is a lengthy process and thus needed to be done ahead of the campaign. However, the decision which targets would be flown on a given day was only made each morning, based on the local weather conditions at the desired target sites. Thus, a spontaneous realignment of the flight direction according to the sun position was not feasible on such short notice. Some separate post-processing techniques to mitigate such artifacts have been proposed by i.e., Queally et al. (2022); Greenberg et al. (2022); Wang and Liu (2016), but have not been tested on the MACS datasets. Depending on a user's requirement and their desired application, some algorithms might be more suitable than others."*

**14. Page 24, line 419. The statement may not be accurate, as there could be inherent errors associated with onboard GPS positioning itself.**

Generally, we agree with the referee that a vertical offset can be attributed to the GPS positioning errors. However, this error affects all images in the dataset more or less equally. In this section, we highlight the sources for the larger-than-average altitude offset at the border of two flightlines that have been acquired with a longer time shift in between. This time shift, coupled with the low across-track overlap of the data acquired over the TrailValleyCreek target have resulted in these more extreme elevation mismatches. We have realized that the formulation of the paragraph and the location within the manuscript were not obvious in this regard. We have thus reformulated some sentences and moved the paragraph to Section *5.2 Changing illumination*.

Added on page 28, line 500: "[...] for further processing. *While almost all targets were flown in a regular grid pattern, where neighboring flightlines were acquired directly one after the other (resulting in time shifts of ca. 10 to 15 min between neighboring images), the TrailValleyCreek target was flown in larger loops. Figure 13e shows the order and flight direction of the lines for the aerial grid. Between the acquisitions of flightlines 2 and 17, approximately three hours have passed and illumination brightness and angle changed strongly. For this extreme case, we conducted a comparison of one of our TVC DSM subsets (Fig. 13b) with the airborne laser scanning (ALS) digital terrain model (DTM, Fig. 13c) that was acquired during the same flight (Lange et al., 2021a). The ALS DTM is of high quality with an accuracy of 0.03 m and a precision of 0.08 m. We applied the DEMcoreg algorithm*

*(Shean et al., 2016) which is based on the method outlined in Nuth and Kääb (2011) to first align the MACS DSM to the ALS DTM and then conduct differencing. We found a vertical offset of -1.40 m and a horizontal offset of -0.03 m in x-direction and -3.44 m in y-direction (see Fig. 13). As the flights over TVC were also flown at only 20 % across-track overlap, this mismatch could not be corrected from further outward-lying, neighboring flightlines (i.e., 4 and 15)."*

**Technical corrections:**

Thank you for noticing these! We have corrected all spellings, grammar errors, and figures accordingly.

**15. Page 2, lines 29-32. The sentence "In addition, …, in the permafrost region." appears somewhat lengthy. It is recommended to split it into two sentences to clarify the cause-and-effect relationship.**

Added on page 2, line 32: *"In addition, the thawing ground strongly affects the lives and livelihoods of the communities in permafrost regions: Anthropogenic infrastructure built on and into permafrost like roads, houses, and pipelines is increasingly at risk, maintenance costs are rising strongly, damages have led to loss of economic value, and the risk of exposure to environmental hazards from thawing grounds has increased as well [...]."*

**16. Page 5, line 98. "The mean annual air temperatures 1990-2020 were …" should be "The mean annual air temperatures for 1990-2020 were …".**

Added on page 5, line 104: "The mean annual air temperatures *for* 1990-2020 were [...]"

**17. Page 6, line 132. In the sentence "50 to 90% permafrost coverage": The expression "50" is not properly formatted and should be written as "50%" to avoid potential ambiguity. "50" and "50%" represent two different numerical values.**

Added on page 6, line 138: "[...] (50 *%* to 90 % permafrost coverage)."

**18. Page 7, Figure 2. In the title: "… the two right sensors the RGB …" should be "… the two right sensors are the RGB …".**

Page 8, figure 2: We updated the image and caption and changed this sentence entirely.

**19. Page 12, line 261. Where is Sec. A? Appendix?**

Added on page 14, line 311: "A more detailed analysis can be found in *Appendix A.*"

**20. Page 13, line 286. The order of letters within the parentheses is incorrect. It should be (B-G-R-NIR) instead of the current sequence.**

Added on page 15, line 337: "[...] blue - green - red - near-infrared *(B-G-R-NIR)* [...]"

**21. Page 19. The page number obstructs the main text.**

Pages 20-22, figures 7-9: We have adapted the image size.

**22. Page 23, Figure 11. In the title: The numbering of subfigures is incorrect. It should be (a) and (b), (c) and (d)...**

Page 27, figure 12: "(a) and *(b)* show RGB orthomosaics generated from images with strong illumination differences between neighboring flight lines. *(c)* and (d) show the corresponding digital surface models (DSMs) of (a) and *(b)* respectively."

**23. Page 24, lines 410-411. "where" should be "were".**

Added on page 28, line 500: "While almost all targets *were* flown in a regular grid pattern, [...]."

**24. Page 35. The page number obstructs the main text.**

Page 34, figure A1: We have adapted the image size.

Community Comment #1:

**This paper presents airborne imagery data and associated products processed from them, acquired over a several year period in the Arctic. It is clear that a tremendous amount of work and expense went into the collection of these data and that they will be useful in a wide variety of studies.**

We thank the reviewer for his statement on the usefulness of this extensive dataset and his acknowledgement of the large amount of work and expense that went into acquiring the data.

**1. However, the paper itself falls short of the mark for ESSD's requirements and I recommend publication only after substantial revisions. That being said, I do not think it will take much work to revise the paper and my comments here are suggestions to the authors to create a paper that will cast the widest net possible to convince others to use and get the most out of their data.**
**In broad brush strokes what needs to be greatly improved is:**
**        1) The description of the photogrammetric system**
**        2) The description of the acquisition flight planning choices**
**        3) The description of the data's accuracy and precision**

We agree with the reviewer that a more detailed description of the photogrammetric system, flight planning choices, and data characteristics will be helpful for readers and will enhance the manuscript and the future usability of the datasets.
We accordingly added relevant information in the manuscript and expanded on the 3 main points raised by the reviewer. Please see detailed comments further below on how specific revisions were implemented.

**2. Some other sections are perhaps over-described, but these comments are not as critical. For instance, there is a comprehensive literature review of permafrost topics which seems to have little bearing on the rest of the paper – either this should be reduced or later the paper should elaborate in more detail how this literature review affected their SPECIFIC flight planning and future science questions. For example, were there acquisitions specifically designed to look at lake drainages, ice wedge melt, beaver ponds, etc., and what questions will these data help answer? If so, which PARTICULAR flight blocks align with which topic?**

Since our datasets were acquired with the specific goal in mind to study permafrost landscapes and their various changes, we see it as important to set the stage with a short review of the pertinent permafrost literature and the reasons why different scientific fields are interested in studying permafrost. The overall length of this permafrost background review in the introduction is 25 lines of text, followed by introducing the topic of remote sensing for permafrost studies, which equally is critical for this paper. This general review does not need to affect the specific flight planning as it forms the broad justification for the need of remotely sensed permafrost landscape data acquisitions. We agree that we need to elaborate in more detail how and why the acquisitions were designed the way they were. We have added some more information in the Conclusion section. We have also added a Table A1, containing some keywords of frequent and/or important features that are present in the

respective datasets. To consolidate dataset detection, we have also added a "reverse" of Table A1, where we describe the used keywords in more detail and collect all the dataset IDs that contain the feature in question (see Table A2). See also replies to comments #20, #31, and #47.

Added in line 551, page 31: *"Potential research with these data sets may include, but is not limited to: tracking coastal erosion (e.g., impending block failures); the detailed analysis of ice-wedge polygons, their microtopography, and their degradation dynamics; monitoring thaw subsidence to evaluate potential impacts on infrastructure; the detection and characterization of retrogressive thaw slump and thermo-erosion gully dynamics; the detailed analysis of ground characteristics in recent and historic fire scars; the detection of lake drainages and drainage pathways in thermokarst lakes (Jones et al., 2023); the examination of individual shrubs and trees in the shrub-tundra regions; or the quantification of beaver dams and lodges (Fig. 17)*
*Given the absence of radiometric calibration targets, calculating radiometric indices may not be wholly reliable with the current version. However, it is important to note that these constraints do not diminish the datasets' utility for object-based analyses, automated segmentation tasks or the mapping of specific features, their distribution, and their microtopography."*

Added Table A2, page 37:

**Table A2.** Overview of keywords including detailed explanation and list of dataset IDs (see Tables 2 and A1) connected with this feature.

| Keyword | Description | Found in the following datasets |
|---|---|---|
| beaver | signs of beaver activities, e.g., dams, lodges, etc. | 18, 20, 25, 31, 32 |
| coast | any type of marine coast; including beaches, coastal bluffs, artificial embankments (in settlements), etc. | 1, 4, 10, 12-16, 19, 20, 28-30, 32, 34 |
| delta | rivers discharging into the ocean, lakes in the form of deltas | 4, 11, 20, 34 |
| drained lake | all types of recent and old drained lake basins independent of cause (including catastrophic drainage, drying, or terrestrialization) | 2, 4-6, 8-11, 14-18, 20, 22, 23, 25, 28, 30- 33 |
| erosion | erosion along marine coasts, lake, and river shores | 1-4, 10, 12, 14-16, 19-23, 25, 26, 29, 33 |
| fire | fire scars with impacts to the landscape still visible in the imagery | 5, 7, 8, 17, 24, 25 |
| forest | mostly groves, some forests. Not applied for isolated trees | 3, 21, 23, 26 |
| gully | thermokarst erosion gullies | 1-3, 17, 20, 23, 25-28, 30-33, 35 |
| ice-wedge polygons | landscapes characterized by high- or low-centered ice-wedge polygons | 2-11, 13-18, 20-26, 28, 30-33, 35 |
| infrastructure | villages, roads, ports, airports, landing strips, landfills, bridges, etc. | 2, 3, 13-16, 19, 21, 22, 29, 33 |
| lake | lakes and ponds >100 m$^2$ (i.e., trough/polygon ponds are not included) | 2-4, 6-11, 13-16, 18, 20-22, 24-26, 28, 30-35 |
| LTO | long-term observatory sites | 1, 3 |
| pingo | pingos | 4, 25, 30-33 |
| river | rivers of any sizes | 3-5, 7-9, 11, 14-18, 20-23, 25, 26, 30-35 |
| settlement | villages and towns | 13, 19, 21, 22, 29 |
| snow | landscapes with snow patches of any size | 7, 26, 32 |
| thaw slump | retrogressive thaw slumps; mostly along coasts and rivers | 1, 14-16, 20, 23, 26, 28, 32 |
| tundra | open tundra landscapes | 1-12, 14-18, 20, 22-28, 30-33, 35 |
| volcanic field | area with volcanic deposits | 35 |

Added Table A1, page 36:

**Table A1.** Overview of all published target areas with information on dataset coverage and resolution, size of the datasets, and flight survey parameters. Further information on features that can be found in the respective datasets, as well as a collection of both literature that informed target selection (published before the acquisition date) and later literature that was conducted in the surveyed area (published after the acquisition date). Extension of Table 2.

| ID | Target | Region* | Date | Area [km²] | GSD [cm] | RGB PC dens [pts/m²] | NIR PC dens [pts/m²] | tiles | sub-projects | # raw images | flight lines | x-track overlap [%]† | Keywords |
|---|---|---|---|---|---|---|---|---|---|---|---|---|---|
| 1 | HerschelIslandEast | WC | 2018-08-15 | 1.38 | 7 | 41.97 | 15.36 | 17 | 1 | 994 | 1 | 2 | 45 | coast, erosion, gully, LTO, thaw slump, tundra |
| 2 | TukRoadGrid | WC | 2018-08-29 | 15.03 | 10 | 9.80 | 3.68 | 78 | 1 | 3263 | 3 + 3‡ | 45 | drained lake, erosion, gully, infrastructure, ice-wedge polygons, lake, tundra |
| 3 | TrailValleyCreek | WC | 2018-08-22 | 161.12 | 10 | 13.88 | 4.86 | 864 | 8 | 42395 | 19 + 4‡ | 28 | erosion, forest, gully, infrastructure, ice-wedge polygons, lake, LTO, river, tundra |
| 4 | CapeSimpson | NA | 2019-07-19 | 23.92 | 7 | 27.82 | 10.49 | 262 | 5 | 22097 | 8 | 45 | coast, delta, drained lake, erosion, ice-wedge polygons, lake, pingo, river, tundra |
| 5 | AnaktuvukRiverFire | NA | 2019-07-22 | 34.94 | 7 | 24.15 | 7.94 | 391 | 5 | 25538 | 12 | 45 | drained lake, fire, ice-wedge polygons, river, tundra |
| 6 | TeshekpukLakeNorth | NA | 2019-07-23 | 107.68 | 7 | 15.11 | 6.40 | 1046 | 11 | 76296 | 23 | 45 | drained lake, ice-wedge polygons, lake, tundra |
| 7 | KetikFire | NA | 2019-07-27 | 72.09 | 7 | 36.15 | 12.59 | 760 | 9 | 45126 | 9 | 45 | fire, ice-wedge polygons, lake, river, snow, tundra |
| 8 | MeadeFire | NA | 2019-07-29 | 48.28 | 10 | 9.56 | 3.32 | 243 | 2 | 8236 | 5 | 45 | drained lake, fire, ice-wedge polygons, lake, river, tundra |
| 9 | NorthSlopeCentral | NA | 2019-07-29 | 54.72 | 10 | 8.86 | 3.08 | 295 | 4 | 11346 | 5 | 45 | drained lake, ice-wedge polygons, lake, river, tundra |
| 10 | DrewPoint | NA | 2019-07-30 | 104.51 | 10 | 7.26 | 3.00 | 406 | 5 | 18415 | 10 | 45 | coast, drained lake, erosion, ice-wedge polygons, lake, tundra |
| 11 | IkpikpukDelta | NA | 2019-07-31 | 13.33 | 7 | 18.40 | 7.46 | 121 | 1 | 8232 | 11 | 45 | delta, drained lake, ice-wedge polygons, lake, river, tundra |
| 12 | ChamisoIsland | WA | 2021-06-25 | 4.05 | 10 | 7.11 | 2.21 | 19 | 1 | 1045 | 6 | 60 | coast, erosion, tundra |
| 13 | Kotzebue | WA | 2021-06-25 | 6.77 | 10 | 5.01 | 2.04 | 35 | 1 | 1103 | 2 | 60 | coast, infrastructure, ice-wedge polygons, lake, settlement |
| 14 | CapeBlossom | WA | 2021-06-25 | 22.32 | 20 | 3.24 | 1.23 | 29 | 1 | 2592 | 3 | 60 | coast, drained lake, erosion, infrastructure, ice-wedge polygons, lake, river, thaw slump, tundra |
| 15 | CapeBlossom | WA | 2021-06-25 | 8.97 | 7 | 29.93 | 10.72 | 103 | 1 | 5450 | 4 | 60 | coast, drained lake, erosion, infrastructure, ice-wedge polygons, lake, river, thaw slump, tundra |
| 16 | CapeBlossom | WA | 2021-06-27 | 23.28 | 10 | 6.98 | 2.85 | 114 | 1 | 4025 | 4 | 60 | coast, drained lake, erosion, infrastructure, ice-wedge polygons, lake, river, thaw slump, tundra |
| 17 | BucklandFireScar | WA | 2021-06-27 | 50.82 | 7 | 37.17 | 12.91 | 515 | 6 | 44492 | 18 | 60 | drained lake, fire, gully, ice-wedge polygons, river, tundra |
| 18 | BaldwinPeninsulaNorth | WA | 2021-06-28 | 16.68 | 10 | 7.16 | 2.83 | 90 | 1 | 3194 | 9 | 60 | beaver, drained lake, ice-wedge polygons, lake, river, tundra |
| 19 | Shishmaref | WA | 2021-06-28 | 8.36 | 10 | 10.91 | 3.98 | 47 | 1 | 3626 | 4 | 60 | coast, erosion, infrastructure, settlement |
| 20 | BPSouth | WA | 2021-06-28 | 85.67 | 20 | 4.18 | 1.61 | 116 | 3 | 10862 | 15 | 60 | beaver, coast, delta, drained lake, erosion, gully, ice-wedge polygons, lake, river, thaw slump, tundra |
| 21 | ShungnakKobukVillages | WA | 2021-07-01 | 19.43 | 10 | 5.44 | 2.46 | 117 | 1 | 2674 | 3 | 60 | erosion, forest, infrastructure, ice-wedge polygons, lake, river, settlement |
| 22 | SelawikVillage | WA | 2021-07-01 | 5.37 | 10 | 5.31 | 2.01 | 29 | 1 | 888 | 4 | 60 | drained lake, erosion, infrastructure, ice-wedge polygons, lake, river, settlement, tundra |
| 23 | SelawikSlump | WA | 2021-07-01 | 15.67 | 10 | 5.22 | 2.38 | 78 | 1 | 1516 | 5 | 60 | drained lake, erosion, forest, gully, ice-wedge polygons, river, thaw slump, tundra |
| 24 | NoatakValleyN | WA | 2021-07-02 | 51.04 | 7 | 24.48 | 9.16 | 557 | 4 | 30388 | 7 | 60 | fire, ice-wedge polygons, lake, river, tundra |
| 25 | NoatakValleyS | WA | 2021-07-02 | 120.71 | 20 | 3.18 | 1.15 | 185 | 4 | 13802 | 6 | 60 | beaver, drained lake, erosion, fire, gully, ice-wedge polygons, lake, pingo, river, tundra |
| 26 | NoatakSlump | WA | 2021-07-02 | 4.56 | 10 | 5.76 | 2.12 | 32 | 1 | 564 | 2 | 60 | erosion, forest, gully, ice-wedge polygons, lake, river, snow, thaw slump, tundra |
| 27 | NoatakRiverS | WA | 2021-07-02 | 12.94 | 10 | 6.82 | 2.75 | 78 | 1 | 1568 | 3 | 60 | gully, lake, river, tundra |
| 28 | NoatakCoast | WA | 2021-07-03 | 27.46 | 10 | 10.22 | 3.50 | 138 | 1 | 5515 | 9 | 60 | coast, drained lake, gully, ice-wedge polygons, lake, thaw slump, tundra |
| 29 | Kivalina | WA | 2021-07-03 | 4.14 | 10 | 3.42 | 1.28 | 23 | 1 | 522 | 2 | 60 | coast, erosion, infrastructure, settlement |
| 30 | SPNorthDTLBEast | WA | 2021-07-09 | 22.13 | 10 | 8.97 | 3.15 | 117 | 1 | 3917 | 12 | 60 | coast, drained lake, gully, ice-wedge polygons, lake, pingo, river, tundra |
| 31 | SPNorthDTLBWest | WA | 2021-07-09 | 33.69 | 10 | 9.54 | 3.30 | 179 | 2 | 8365 | 11 | 60 | beaver, drained lake, gully, ice-wedge polygons, lake, pingo, river, tundra |
| 32 | SPNorthKittukCoast | WA | 2021-07-09 | 97.67 | 10 | 9.16 | 3.13 | 462 | 4 | 20098 | 18 | 60 | beaver, coast, drained lake, gully, ice-wedge polygons, lake, pingo, river, snow, thaw slump, tundra |
| 33 | SPCKougarok01 | WA | 2021-07-10 | 109.45 | 10 | 8.08 | 2.87 | 549 | 5 | 18123 | 15 | 60 | drained lake, erosion, gully, infrastructure, ice-wedge polygons, lake, pingo, river, tundra |
| 34 | KobukDelta | WA | 2021-07-10 | 84.14 | 20 | 0.78 | 1.20 | 108 | 2 | 12081 | 10 | 60 | coast, delta, lake, river |
| 35 | SPCInuruk | WA | 2021-07-10 | 84.72 | 10 | 12.06 | 4.36 | 454 | 5 | 30299 | 20 | 60 | gully, ice-wedge polygons, lake, river, tundra, volcanic lava field |

* WC: West Canada, NA: North Alaska, WA: West Alaska; † this number corresponds to the planned and targeted across-track overlap for grid flights. As detailed in Sec. 2.3, the actual overlap can deviate in the resulting data;‡ 3 (19) flightlines were acquired in the main grid direction, 3 (4) further lines were acquired roughly perpendicular to the first 3 (19).

**3. Similarly, there was a tremendous amount of detail on the image processing steps such as vignetting – is there a reason these steps can't be reduced to a single sentence? That is, was there something unique about this processing or will the information provided be important to someone using the data? Etc.**

We believe that the detailed image processing descriptions of our specific imagery data are important and useful to readers and future users. Since we are not using standard prosumer cameras such as a Nikon D800 but a custom-made Modular Aerial Camera System (MACS) designed and fabricated by the German Aerospace Center's Institute of Optical Sensor Systems, the details on the image pre-processing are important in our view. The subsection on 'Devignetting and file format conversion' is 14 lines long, with only three sentences on devignetting. If the desire to shorten this paragraph is shared by the editorial office, we could suggest removing Figure 4. However, as the Figure shows the effects of our devignetting process and thus the effects on enhancing our specific imagery, we would prefer keeping it here.

**4. The term 'super-high-resolution' is used in the title and throughout the paper and this needs to be changed. What is 'super-high' to you may be coarse to someone else. Especially when it comes to modern airborne photogrammetry, there is nothing 'super-high' about 10 cm GSD.**

We agree with the reviewer that the term "super"-high-resolution might be somewhat relative from a reader's stand point or experience. While the resolution offered in our datasets clearly surpasses the resolutions offered by very-high-resolution satellite sensors (0.3 - 0.5 m for commercial sensors), we now use the more widely accepted term "very-high-resolution" throughout the manuscript and the title.

We removed the term *"super-high-resolution"* and replaced it with *"very-high-resolution"* throughout the manuscript and the title.

**5. Also, the term resolution is not the best choice in most of these cases, though commonly used. A better choice is GSD, which is used elsewhere in the paper, when talking about the area covered by a single pixel and reserving 'resolution' to discuss whether the shape of an ice wedge or a tussock is resolved or not, though that's a little nitpicky (though not in the title).**

We agree that in the preprint, the terms 'resolution' and 'GSD' have been used interchangeably. We have reviewed all incidences and consolidated the proper uses. For the title however, we kept the term "resolution", as this is what readers intuitively understand when considering the relevance of such a dataset.

Changed one instance of *"GSD"* to *"spatial resolution"*.
Changed 11 instances of *"spatial resolution" / "spatial resolutions" / "image resolutions" / "resolutions"* to *"GSD" / "GSDs"*.

**6. Similarly, I feel the rest of the title is a disservice to the data the authors are presenting – what is described in this paper are blocks of images processed into**

orthomosaics and DEMs, the imagery itself is just an intermediate step in this case. You want people to read the paper and use the processed data products, right? If so, pick a title that will draw savvy users into doing so.

We agree with this observation and have thus adapted the title according to the reviewer's comment to now include the keywords 'orthomosaics', 'point clouds' and 'elevation data'.

New title: *"Very-high-resolution aerial image orthomosaics, point clouds, and elevation datasets of permafrost landscapes in Alaska and northwestern Canada"*

7. So overall I think there is too little detail where detail matters to future users of these data and too much detail on topics that won't be of much to them, and at least the parts with too little details need to be addressed to give these data the longest legs possible. The paper also needs some reorganization, as important information on methods or results is sprinkled into somewhat random locations throughout the text. My review focuses on these broad brush strokes and some science questions/comments I have, as I think it's premature to discuss word choices or section structure though in general the writing is clear and well written so those comments would be few in number any way. I would be happy to re-review this paper or answer any questions that I could in the meantime.

Thank you for the offer to re-review the paper, especially as we see how much time and expertise has gone into the comments so far.
Concerning the structure of the manuscript, we hope that through the multitude of other changes implemented (e.g., comments #2. #24, #25, #31, #47, …), we have managed to consolidate the information that went astray in other sections and can now present a more organized manuscript in this second round. We hope that the information flow is clearer now and users of the datasets can more easily find any information of interest.

**Major Revisions**
1) System Description.
The paper essentially lacks a section on the photogrammetric system itself and this is unacceptable for a data paper on photogrammetric products in ESSD. While it's fine to reference other papers that contain various details, the broad brush strokes MUST be included here if this is the first time this system has been used for this purpose, as seems to be the case. There is a brief section that describes the camera,
but even the camera description is insufficient. Here is what MUST be addressed at MINIMUM in my opinion:

10. Was there a GNSS system installed? If so, give some basics about it. No mention was made that you recognize that the GNSS antenna is not at the camera and that you dealt with the lever arms appropriately in flight direction and crabbing.

Survey-grade aerial cameras regularly do not have the GNSS antenna at the camera. To make this information clear to non-expert users of our dataset as well, we have included information on the lever arms in the revised chapter 2.2 describing the camera system.

Changes made in line 168, page 7: *"Relevant distances between lever arms like the IMU to GNSS antenna and the IMU to sensor are measured with an uncertainty smaller than 10 mm. These offsets are stored on the GNSS receiver and used to calculate the correct sensor position."*

**11. Exactly how was the camera triggered and how was the time of photo capture recorded relative to the GNSS data stream? What is the timing accuracy?**

The timing accuracy is 10 ns which is sufficient for georeferencing the NIR and RGB images among each other. However, this is a very technical aspect and from our point of view of minor relevance for most of the readers.

Changes made in line 164, page 7: *"All sensors are electrically triggered to start the image exposure at exactly the same time. At the end of integration, the sensor delivers a pulse to the GNSS receiver, generating a message including time, position and attitude. This georeference is written into the corresponding raw file of the aerial image before storage and can be later substituted by a post-processed solution."*

**12. What is the resulting spatial accuracy of the photo centers? This CONTROLs the precision and accuracy of a final gridded products so must be stated or referenced.**
**In what ways is this custom camera superior to a Nikon D800 or D850, which were available at the time of these acquisitions and have far superior megapixels and a huge dynamic range?**

We understand the importance of specifying this information as it controls the precision and accuracy of the final gridded products. We revised Chapter 2.2 which now includes additional detail on the camera and GNSS in the revised manuscript. We like to emphasize that the term "superior megapixels" does not say anything about optical resolution, optical stability and geometric accuracy of the overall camera. As these are important in the field of professional remote sensing, prosumer cameras are rarely used. With regards to the comparison between our custom-made industrial aerial camera and the prosumer cameras Nikon D800/D850, we would like to point out that the MACS has 9500 pixels below the aircraft, surpassing the D850's 8300 pixels. Additionally, while both cameras produce 14-bit images, the MACS camera's capabilities can be enhanced by multiple exposure times. As the main focus of our paper shall remain on the described datasets and their processing, we do not include a full comparison of prosumer-grade cameras vs. the system we have used here; especially as the choice of prosumer camera to compare to would be a subjective one. For more information on the technical details of the MACS, please refer to Lehmann et al. (2011).

Changes made in line 155, page 7:
*"**2.2 Multispectral sensor***
*For all three campaigns, we used a custom-built aerial camera system (configuration: MACS-Polar18) developed by the DLR Institute of Optical Sensor Systems. It was specifically adapted to work in very low ambient temperatures. Multiple integration times per scene can be acquired to avoid under- or over-exposed pixels in the challenging Arctic light conditions, where very dark (water, dark bare soil) and bright (snow, ice) surfaces often co-exist in target areas (Brauchle et al., 2015).*

*The camera consists of a computing unit and a sensor unit (sensor: SVS Vistek HR16070CFLGEC). The computing unit comprises sub-assemblies including a L1/L2/L-band GNSS receiver and the main computer. The sensor unit contains an inertial measurement unit (IMU) and three 16 megapixel (MP) industrial cameras: one nadir-looking NIR camera and two visible RGB cameras with overlapping right- and left-looking (+/- 8.5°) view directions (Fig. 2a, b). The maximum frame rate is 4 fps. Thus, when acquiring at two different exposure times, each is repeated with a rate of 2 fps. All sensors are electrically triggered to start the image exposure at the exact same time. At the end of integration, the sensor delivers a pulse to the GNSS receiver, generating a message including information on the time, position and attitude of the acquisition. This georeference is written into the corresponding raw image file before storage and can later easily be substituted by a post-processed solution. Relevant distances between lever arms like the IMU to GNSS antenna and the IMU to sensor are measured with an uncertainty smaller than 10 mm. These offsets are stored on the GNSS receiver and used to calculate the correct sensor position. Technical specifications are summarized in Table 1. More information on the camera system and its general concept can be found in (Lehmann et al., 2011). The MACS-Polar18 was operated onboard the Polar-5 (2018) and the Polar-6 (2019, 2021) polar research aircraft, which represent AWI's own research fleet (Alfred Wegener Institute, 2024). The aircraft are Douglas DC-3 planes refitted to Basler BT-67 planes for harsh polar environments. The sensor unit is installed in a belly port of the planes (Fig. 2c and d)."*

**13. What is the dynamic range of the MACS sensor in EV? What is the focal length of the lens for each sensor? What are the pixel dimensions of the sensor and what are the swath widths for each at a typical GSD?**

Thanks for highlighting that this information was missing; we agree that it might be relevant for some users and professional readers. Accordingly, we extended Table 1 to include information on the dynamic range, the focal length, the pixel size, and the swath width. The swath width is also pictured in Fig. 2b, which shows the footprint of the MACS sensors at 1000 m GSD.

Changes made: Extension of Table 1, page 9:

| | | |
|---|---|---|
| **Frame rate max.** | 4 fps continuous | |
| **Operating temperature min.** | -20 °C | |
| **Weight** | 17 kg | |
| **INS*** | GNSS Novatel OEM6 / IMU Sensonor STIM300 | |
| | **2 × RGB sensor** | **1 × NIR sensor** |
| **Wavelength range (FWHM†)** | B: 450-510 nm
G: 490-580 nm
R: 580-700 nm | 715-950 nm |
| **Dynamic range** | 62 dB | 62 dB |
| **Sensor resolution / pixel pitch** | 16 MP / 7.4 $\mu$m | 16 MP / 7.4 $\mu$m |
| **Focal length** | 90 mm | 50 mm |
| **IFOV‡** | 81 $\mu$rad | 140 $\mu$rad |
| **FOV§ (cross × along track)** | 23° × 16° per sensor
44° x 16° both sensors | 40° × 22° |
| **Swath width (at 1000 m AGL)** | 400 m | 700 m |
| **GSD (at 1000 m AGL)** | 8 cm | 15 cm |

\* inertial navigation system; † full width at half maximum; ‡ instantaneous field of view; § field of view

**16. What camera parameters are fixed and which are set in the air? For example, focus, aperture, shutter speed, iso. What values were used here (in general) and how were they determined or changed in flight? What minimum shutter speed were used in particular and how does this relate to pixel blur caused by the aircraft's motion?**

As we were working with an aerial camera, all parameter settings are fixed - except the exposure time. We agree that this might be interesting to some readers and have added it to the text.

Changes made in line 200, page 9: *"For our aerial camera all parameters are fixed except exposure time, typically ranging from 0.2 ms to 1.5 ms. Given a motion rate of 6.7 cm/ms (aircraft speed was roughly 130 kts), the resulting motion-induced blur at e.g., 1.5 ms exposure time is approximately 1.2 pixels in an 8 cm ground sample distance (GSD) scenario."*

**17. What is a Polar 5 aircraft? Later it is described as a modified DC3 (which I find really cool) but why is such a huge plane required here compared to a more maneuverable aircraft which burns less fuel and more easily makes tight turns for grids or following irregular features like rivers?**

The Polar-5 (flown in 2018) and Polar-6 (flown in 2019 and 2021) aircrafts are part of AWI's polar research infrastructure. The aircrafts are modified DC3s chosen for their unique capabilities that align with our remote sensing objectives in different Polar regions. Despite smaller, more maneuverable aircraft having fuel efficiency and ease of navigation advantages, the Polar-5 and -6's larger size offers stability, higher payload capacity, and transferability across the entire Arctic and Antarctic domain. This ensures a stable platform for our remote sensing equipment in various environments including very remote regions, and can operate different science configurations including optical sensors, LiDARs, meteorologic sensors, greenhouse gas concentration sensors, radar sensors, gravimetric sensors, and more which are routinely flown over different Arctic and Antarctic areas. In this sense, we have favored these aircraft (that are slightly larger than necessary for an airborne imaging campaign), over any other platforms, as they represent fully operable research planes that are directly available to us at our institute.
We do not think that a comprehensive explanation is necessary in this data description paper, but have slightly adapted the text to add the information that these aircraft belong to our institute's own fleet, in case other readers also wonder about this.

Changes made in line 177, page 7: *"The MACS-Polar18 was operated onboard the Polar-5 (2018) and the Polar-6 (2019, 2021) polar research aircraft, which represent AWI's own research fleet (Alfred Wegener Institute, 2024). The aircraft are Douglas DC-3 planes refitted to Basler BT-67 planes for harsh polar environments. The sensor unit is installed in a belly port of the planes (Fig. 2c and d)"*

**18. Were there other sensors installed that required the room? Were any of these sensors turned on at the time of the photogrammetric acquisitions and thus have some utility to users of the photogrammetric data? Was the primary mission of these flights to do photogrammetry or something else?**

Typically, multiple sensors are actively collecting data during our flight campaigns. For a comprehensive overview of additional sensors and the data they generate, we recommend referring to the "Data Availability" section, where we have provided detailed information on their specifications and the data they collect (see page 32, lines 570):
"During all three campaigns described in this publication, additional sensors, i.e. a full-waveform LiDAR (Riegl LMS-Q680i), a slewable camera, an infrared thermometer, as well as sensors for measuring air temperature, moisture, and barometric pressure were installed on the planes and recorded measurements at the same time as the MACS recorded images. For the flights in North Alaska in 2019, a further sensor recorded methane concentration. Data of these additional sensors has not been published so far and is available from the authors upon reasonable request.". As the majority of these datasets are not published yet, we cannot provide links to any repositories at this point. However, as we acknowledge that some users might be very interested in correlating the aerial datasets from this manuscript with additional environmental datasets, we are happy to get in touch with those interested. We have also added a reference to the Data availability section earlier in the manuscript (Sec. 2 Data acquisition).

Changes made in line 77, page 3: *"Aside from this aerial camera system, further environmental sensors acquired data during survey flights. We do not report on them in this publication, but some further information on their availability can be found in Sec. 7."*

**1) Flight Planning.**
**19. The section labeled Survey Design does not adequately describe why flight parameter decisions were made and these are important given the unusual choices that were described. As I understand the data described here (which I have not attempted to download), the authors are only treating the data that were acquired in blocks. Yet, little information is provided about these blocks. Figure 1 sort of shows their general location but this is largely obscured by the black lines which are apparently irrelevant to this paper and by the large spatial scale. Figure 1 needs to be revised to show only the blocks (in a, b, c) with enough scale to see exactly where they are and how many lines are within each block (or annotate that). As it appears at this scale, most of these blocks are only two passes? If true, this is important to know.**

We understand that Fig. 1 is not ideal to get an all-encompassing overview with details on all datasets that we flew and publish here. As we have covered targets spread over large spatial domains, adding maps that would provide these details (especially the number of flightlines) would require multiple page-filling figures. In our opinion this would disrupt the flow of the manuscript too much, which is why we had decided to only give a general overview in Fig. 1. Nevertheless, we provide the footprints of all our flights and those of all published datasets as a geopackage. We have added a reference to these footprint files both in the caption of Fig. 1 and in the Section 4 Data and metadata structure.

We have also revised Fig. 1 in general: We have updated the colors to make the contrast stronger between the footprints of all acquired images (visible in black in Fig. 1d; only very faint in Fig. 1a-c) vs. the footprints of the images that make up the published datasets from this publication (pink in Fig. 1a-c). We have further labeled the targets based on the target ID from the first column in Tab. 2 (which has also been added in the revised manuscript).

The information, how many flightlines went into which target has also been added to Tab. 2. With both these additions/revisions, more information can be drawn by combining Fig. 1 and Tab. 2.

Changes made: *see comment #20*

**20. Further, the text (in this section and in 2.1) describes a variety of great reasons that drove flight-planning decision-making, but there is nothing I found that relates back to specific blocks presented here – the blocks should be color coded or otherwise annotated to refer to their relevance according to scientific driver and the text limited and focused to only those scientific topics actually covered by the data here, if you want to entice others to make the most use of them. And better yet, the references should relate to the blocks too – if you mapped an area specifically because some paper noted something of scientific significance occurring there, this should be made clear to the reader who may be interested in that topic or area so that they are motivated to find your data. For example, did you map any beaver dams? Or fire scars? Etc And which blocks were those? And any villages mapped as blocks need to be identified on the figures.**

Color coding the blocks shown in Fig. 1 to highlight the specific scientific purposes and flight planning decisions does not help with readability in our opinion. Especially since image blocks contain a wide variety of different features. Instead, we have added the IDs to Fig. 1 to allow for cross-referencing with Table 2 and Table A1. Table 2 now shows a short version with information on dataset essentials (acquisition region, date, area covered), and information on spatial resolution. Table A1 is an extended version of Table 2 and can be found in the Appendix with additional information, including the number of flightlines (see comment #19), as well as some keywords/tags that help categorize the type of data/information/features that can be anticipated within the data. However, we need to stress that this list is not guaranteed to be exhaustive and that potential users may likely have different intended uses than we might have planned for our research. We did not add any mapped villages to the maps, to avoid further cluttering the figure. But we have added a keyword "settlement" in the corresponding column of Table A1.

Changes made:
Updated Table 2:

**Table 2.** Overview of all published target areas with information on acquisition parameters, dataset coverage and resolution. An extended version can be found in the Appendix, Table A1.

| ID | Target name | Region* | Date | Area [km²] | GSD [cm] | RGB PC dens [pts/m²] | NIR PC dens [pts/m²] |
|----|-------------|---------|------|-----------|----------|---------------------|---------------------|
| 1 | HerschelIslandEast | WC | 2018-08-15 | 1.38 | 7 | 48.03 | 17.85 |
| 2 | TukRoadGrid | WC | 2018-08-29 | 15.03 | 10 | 10.28 | 3.86 |
| 3 | TrailValleyCreek | WC | 2018-08-22 | 161.12 | 10 | 14.29 | 4.66 |
| 4 | CapeSimpson | NA | 2019-07-19 | 23.92 | 7 | 28.17 | 10.17 |
| 5 | AnaktuvukRiverFire | NA | 2019-07-22 | 34.94 | 7 | 23.94 | 8.32 |
| 6 | TeshekpukLakeNorth | NA | 2019-07-23 | 107.68 | 7 | 17.52 | 6.44 |
| 7 | KetikFire | NA | 2019-07-27 | 81.35 | 7 | 36.27 | 12.23 |
| 8 | MeadeFire | NA | 2019-07-29 | 52.73 | 10 | 9.46 | 3.32 |
| 9 | NorthSlopeCentral | NA | 2019-07-29 | 58.56 | 10 | 8.82 | 3.08 |
| 10 | DrewPoint | NA | 2019-07-30 | 104.51 | 10 | 8.15 | 3.05 |
| 11 | IkpikpukDelta | NA | 2019-07-31 | 14.16 | 7 | 22.73 | 8.83 |
| 12 | ChamissoIsland | WA | 2021-06-25 | 7.44 | 10 | 24.44 | 8.36 |
| 13 | Kotzebue | WA | 2021-06-25 | 12.11 | 10 | 5.75 | 2.01 |
| 14 | CapeBlossom | WA | 2021-06-25 | 25.22 | 20 | 3.66 | 1.34 |
| 15 | CapeBlossom | WA | 2021-06-25 | 9.62 | 7 | 29.12 | 10.33 |
| 16 | CapeBlossom | WA | 2021-06-27 | 23.28 | 10 | 7.55 | 2.86 |
| 17 | BucklandFireScar | WA | 2021-06-27 | 50.82 | 7 | 36.87 | 12.73 |
| 18 | BaldwinPeninsulaNorth | WA | 2021-06-28 | 16.68 | 10 | 8.43 | 2.94 |
| 19 | Shishmaref | WA | 2021-06-28 | 9.52 | 10 | 11.32 | 4.27 |
| 20 | BPSouth | WA | 2021-06-28 | 85.67 | 20 | 4.20 | 1.51 |
| 21 | ShungnakKobukVillages | WA | 2021-07-01 | 20.87 | 10 | 5.91 | 2.45 |
| 22 | SelawikVillage | WA | 2021-07-01 | 5.74 | 10 | 5.81 | 2.35 |
| 23 | SelawikSlump | WA | 2021-07-01 | 15.67 | 10 | 5.76 | 2.34 |
| 24 | NoatakValleyN | WA | 2021-07-02 | 51.04 | 7 | 26.72 | 9.17 |
| 25 | NoatakValleyS | WA | 2021-07-02 | 120.71 | 20 | 3.26 | 1.18 |
| 26 | NoatakSlump | WA | 2021-07-02 | 5.21 | 10 | 5.19 | 2.07 |
| 27 | NoatakRiverS | WA | 2021-07-02 | 12.94 | 10 | 7.30 | 2.75 |
| 28 | NoatakCoast | WA | 2021-07-03 | 27.46 | 10 | 10.21 | 3.58 |
| 29 | Kivalina | WA | 2021-07-03 | 4.14 | 10 | 3.95 | 1.46 |
| 30 | SPNorthDTLBEast | WA | 2021-07-09 | 22.13 | 10 | 8.95 | 3.16 |
| 31 | SPNorthDTLBWest | WA | 2021-07-09 | 33.69 | 10 | 8.88 | 3.11 |
| 32 | SPNorthKitlukCoast | WA | 2021-07-09 | 97.67 | 10 | 9.38 | 3.17 |
| 33 | SPCKougarok01 | WA | 2021-07-10 | 109.45 | 10 | 8.18 | 2.88 |
| 34 | KobukDelta | WA | 2021-07-10 | 84.14 | 20 | 2.57 | 1.19 |
| 35 | SPCImuruk | WA | 2021-07-10 | 84.72 | 10 | 11.21 | 4.00 |

* WC: West Canada, NA: North Alaska, WA: West Alaska

Updated Table A1:  *see comment #2*
Updated Fig. 1, page 4:

[Figure]

**21. In terms of flight planning, no information was given on the choice of side lap. Why were these sidelaps chosen? Do you believe there was some photogrammetric advantage to using 28% rather than 60%? If this paper and the products described here were essentially opportunistic (that is, the flights were flown for other reasons than creating these data products) that's fine, but this needs to be stated clearly to make clear you are not proposing something non-standard as being superior. No information that I could find indicates how many flight lines composed each block or how accurately you believe they were flown.**

No, we do not think that flying with only 28% across-track overlap has a photogrammetric advantage over higher overlaps such as 60%. Finding the ideal flight parameters (including good trade-offs between quality and efficiency) has been an iterative process. This means that in later campaigns, we had the opportunity to optimize flight planning and learn from our past campaigns. Hence, the overlap was lower in grids from 2018 and 2019 (45%), but increased to 60% for the 2021 campaign. Only one target (TrailValleyCreek) was flown at merely 28% across-track overlap. We have added the overlaps in a new column for Tab. 2 and have adapted the text to make the decision-drivers for the parameters clearer.

Changes made in line 203, page 9: *"The along-track overlap between single image captures is 80 % for all datasets. For targets flown in grid-mode, the across-track overlap is 45 % for datasets from 2018 and 2019. For the campaign in 2021, we have increased the across-track overlap to 60 %. Only one grid (TrailValleyCreek) from 2018 was flown with a side-overlap of only 28 %. The main aim of this flight grid was the acquisition of LiDAR data and thus flightline planning was optimized towards ALS requirements. This led to a significantly lower overlap for the aerial images, which were only a byproduct during this flight. For the photogrammetric processing, the overlaps of 80 % along-track and 45 % to 60 % lead to every ground location being captured 10 to 15 images, and thus from 10 to 15 slightly different viewing angles. These numbers decline towards the edges of grids and can further vary with deviant angles of roll, pitch, and yaw of the aircraft induced by internal (i.e., pilot) or external (i.e., wind, drift) influences."*

Changes made to Tab. 2: *added column "x-track overlap", see also comments #19 and #20*

**22. It's also unclear what the relevance of 'viewing angles' is, what we really need to know is how many image pairs cover each pixel. For 60% sidelap and 80% overlap, this should be 8-10. This places strong controls on precision. But we also don't know the focal length, and this controls the base-height ratio and thus also controls accuracy.**

With 60% across-track and 80% along-track overlap, each pixel should ideally be covered in 15 images. For 28% and 45% across-track overlap, this number declines to 10 images.

Changes made: *see comment #21.*
*The focal length has been added to Tab. 1, see also comment #13.*

**23. When flying grids, did you attempt to maintain a constant AGL? Or was this averaged? How did you determine the flying height AGL in mission planning and how did you maintain it while flying?**

Thank you for pointing this out. We realize that we have failed to address this in the original manuscript. During flights, we maintained a stable altitude ASL and did not follow a constant altitude AGL. We have now added this information to Sec. 2.3 Survey design.

Changes made in line 194, page 8: *"For every flight, the flying altitude was set above sea level (ASL) - determined by adding an offset of 500 m / 1000 m / 1500 m to the ground elevation (depending on the desired resulting GSD of output images) - and remained constant throughout a grid. As most targets show only minimal elevation changes throughout their area, the constant altitude ASL corresponded to an almost constant altitude AGL. For targets that did show differences in elevation, we added the respective offset of 500 m / 1000 m / 1500 m to the higher elevations. With this approach, we made sure to avoid holes in the flown grid (at the cost of lower spatial resolution for the lower-lying areas of the target)."*

**24. There seems to be a variety of information related to flight planning sprinkled throughout the remaining text – this needs to be consolidated here so that a savvy reader has all the information they need in a single spot.**

We have reviewed the manuscript and consolidated any information on flight planning or survey design to Section 2.3 Survey design.

Changes made:
- Added in line 194, page 8: *"For every flight, the flying altitude was set above sea level (ASL) - determined by adding an offset of 500 m / 1000 m / 1500 m to the ground elevation (depending on the desired resulting GSD of output images) - and remained constant throughout a grid. As most targets show only minimal elevation changes throughout their area, the constant altitude ASL corresponded to a constant altitude AGL. For targets that did show differences in elevation, we added the respective offset of 500 m / 1000 m / 1500 m to the higher elevations. With this approach, we made sure to avoid holes in the flown grid (at the cost of lower spatial resolution for the lower-lying areas of the target)."*
- Moved to line 223, page 10: *"Prior to any acquisition flight, we set the camera parameters according to the prevailing illumination conditions. Heavy cloud cover with homogeneous diffuse light, for example, required longer sensor exposure times than clear skies and direct sunlight."*
- Moved to line 225, page 10: *"During some of the flights, we chose different shutter timings for the NIR and the RGB sensors (e.g., 0.2 ms and 0.4 ms respectively). The timing for the slightly more sensible NIR sensor was set a little lower to avoid overexposure in its data. For targets where we expected to see both very bright and very dark ground features (e.g., snow and water bodies), we also acquired images at two different exposure times (e.g., 0.4 ms and 1.0 ms), to ensure that we always had at least one image that was not over- or underexposed. This was also done when we expected shifts in illumination throughout the survey time."*
- Added in line 388, page 19: *"[...] (see also Sect. 2.3)."*
- Added in line 485, page 27: *"Despite our best efforts to optimize the sensor's parameters according to the prevailing light conditions (see Sect. 2.3), [...]"*

**3) Data quality.**
**25. In my opinion, there is simply no useful data quality information here at all and this MUST be addressed. The authors state that many of their locations were selected due to the availability of prior data at these locations, yet there are no comparisons to these data for data quality purposes. Why not? There is not even a reference I could find to prior studies of data quality using this system.**

Thank you for highlighting this. We agree that we have failed to provide a quantitative comparison of the datasets' accuracies in the first version of the manuscript. As you also pointed out in comment #29, the "horizontal accuracy should be within [...] 50 cm at most" and comment #33 "Using modern PPP processing [...] you should be achieving < 10 cm positioning". We have now used the GNSS-corrected nav-files and reprocessed all target sites.
As this study marks the first comprehensive paper using this particular system, there are no other references to prior studies we can make here. However, we have compared horizontal

accuracy of this newly-processed dataset version to imagery published by the NGS / NOAA from the year 2017 and to high-resolution satellite imagery from the year 2020 published by the Alaska Geospatial Office. Horizontal accuracy was also tested with DPGS data collected in the City of Kotzebue. For the vertical accuracy, we have compared the DSMs of these two targets to a LiDAR elevation model (for the AnaktuvukRiverFire target) and to the other DSMs of the same target but at different GSDs (CapeBlossom at 7, 10, and 20 cm GSD). The choice to do the positional accuracy assessment with these two targets was made based on the availability of comparison datasets from the same date (the three datasets of CapeBlossom compared to each other - as wished for in CC #26 - and the AnaktuvukRiverFire as we had already obtained a processed LiDAR DEM at the highest possible GSD of 1 m from this flight from a previous study).

We have now reorganized Section **5 Data quality assessment** and included a subsection describing the methodological approach of the quantitative accuracy assessment and the respective results (see Sect. 5.1). We have reorganized the previously conducted qualitative assessment into Sect. 5.2.

Changes made: "*5.1 Quantitative assessment on geolocational accuracy*

*In order to assess the geolocational accuracy of our processed datasets, we have conducted comparison studies both for the horizontal as well as the vertical precision. We have selected three exemplary datasets within the targets of Cape Blossom, the Anaktuvuk River Fire and the town of Kotzebue (see Table 2, IDs 14–16, 5, and 13).*

*5.1.1 Horizontal precision*

*To determine the precision of our Cape Blossom datasets, we compared our mosaics at all three GSDs (7 cm, 10 cm, 20 cm) towards each other as well as to reference imagery published by the NOAA Office for Coastal Management from 2017. This data is available at 0.35 m GSD with a horizontal positional accuracy of circa 1.5 m and 95% circular error confidence level (NOAA, 2024). For the 2019 data from Anaktuvuk, we compared the accuracy to the Alaska High Resolution Imagery (AHRI) published by the USGS (Maxar Technologies Inc., Alaska Geospatial Office, USGS, 2020). This data is based on Maxar imagery from 2020 and ships at 50 cm GSD with a reported horizontal accuracy of 0.5 m and 95% circular error confidence level. We manually identified tie points within our MACS mosaics and the reference imagery datasets and calculated their offsets to each other in the X-Y-plane. For Cape Blossom, we determined an offset of up to 1.70 m ± 0.29 m (mean of residuals ± std of residuals) towards the NOAA imagery, based on 9 evenly spread tie point locations. Towards each other, the datasets show a maximum shift of 0.28 m ± 0.12 m (Table 3).For the Anaktuvuk imagery we found that the horizontal positioning accuracy of our datasets is within 1.65 m ± 0.10 m compared to the AHRI images, based on 13 evenly spread tie point locations. Considering the reported uncertainties for the NOAA and AHRI imagery, more precise values are not possible. For the Kotzebue mosaics collected on 25 June 2021, we used DGPS measurements of 11 GCPs from 7 July 2021. We divided the GCPs into six inner and five outer points. The inner points are covered by data flown in north-south, as well as in east-west direction, while the outer points are only covered by one direction. Figure 10 marks their locations within the Kotzebue dataset. For the inner GCPs, we measured an X-Y offset of 0.13 m ± 0.02 m, for the outer GCPs, we measured an offset of 0.62 m ± 0.85 m.*

[Figure]

**Figure 10.** Location of six inner and five outer GCPs measured by DGPS in the village of Kotzebue.

**5.1.2 Vertical precision**

*For quantifying the vertical precision, we again compared the Cape Blossom DSM datasets at the different GSDs towards each other. Here, we found average offsets less than 0.60 m from each other, with maximum standard deviations of 0.10 m (Table 3). For the Anaktuvuk River Fire DSM, we had access to a LiDAR-derived elevation model that was collected during the same flight. The average density of the LiDAR point clouds was 5 pts/m2. The resulting elevation model had a GSD of 1 m; the vertical accuracy reached 0.10 m. The process to derive the DEM from the raw data involved classifying target waveforms into vegetation and ground returns, post-processing ground returns into georeferenced point cloud data, and finally constructing the DEM using inverse distance weighting interpolation (see also Rettelbach et al. (2021)). Within the overlapping areas of the processed LiDAR DEM and our photogrammetrically-derived DSM (covering 0.63 km2), we detected a vertical offset of 0.18 m ± 0.06 m between the two elevation models.*

**Table 3.** Horizontal (X-Y) and vertical (Z) offsets between Cape Blossom datasets at different GSDs towards each other and horizontal offsets towards NOAA reference imagery. All values in mean of residuals [m] ± std of residuals [m].

|  |  | MACS 7 cm GSD | MACS 10 cm GSD | MACS 20 cm GSD | NOAA 35 cm GSD |
|---|---|---|---|---|---|
| **MACS 7 cm** | horizontal | — | 0.12 ± 0.08 | 0.28 ± 0.12 | 1.70 ± 0.29 |
|  | vertical | — | 0.06 ± 0.07 | 0.02 ± 0.10 | — |
| **MACS 10 cm** | horizontal | 0.12 ± 0.08 | — | 0.25 ± 0.08 | 1.70 ± 0.29 |
|  | vertical | 0.06 ± 0.07 | — | 0.04 ± 0.09 | — |
| **MACS 20 cm** | horizontal | 0.28 ± 0.12 | 0.25 ± 0.08 | — | 1.58 ± 0.27 |
|  | vertical | 0.02 ± 0.10 | 0.04 ± 0.09 | — | — |

**5.2 Qualitative dataset assessments**

[…]"

**26. Especially given their poor choice of side laps (apparently chosen for lidar purpose?), these photogrammetric DEMs need a rigorous accuracy and precision assessment for each side lap. From section 5.3 I'm surmising that their aircraft was equipped and was using lidar on every flight (???!!!) – if this is true, they have the opportunity to compare EVERY photogrammetrically-derived DEM to their lidar and this should be done if not on all of them then a large subset capturing both flight planning differences and terrain differences. For such small areas, this should only take a day or two total, if that. I mean what's the point of writing this paper and archiving these data if not to be used by others? And how can they be used by others for anything useful without SOME understanding of topographic accuracy and precision?**

We agree with the crucial need for a meticulous evaluation of accuracy and precision (see response to CC #25). While LiDAR data were taken for most areas, ongoing processing demands expertise and additional scrutiny, particularly considering factors like vegetation that may influence the data. Accurate processing of the data will certainly require more than "a day or two". In future and ongoing studies, we aim to advance our comprehension of topographic accuracy and precision for these photogrammetric DEMs and will factor in more LiDAR-derived datasets. For this manuscript however, the focus lies on the provision of the image-derived and photogrammetrically processed datasets acquired by the MACS. It is therefore up to potential users to decide if our free products match their accuracy needs.

The high-resolution LiDAR DEM for the AnaktuvukRiverFire target area that has been processed (for another study) to the highest possible GSD of 1 m with elaborate consideration, has been included for the quantitative assessment of the vertical accuracies (see CC #25).

Changes made: *see CC #25*

**27. Your Figure 3 flowchart does not indicate anything about photo-center geolocation or GNSS interaction – this needs to be updated to make clear how you selected your initial positions for photo centers and in the text stated what you believe the accuracy of those positions are. The accuracy of these photo centers CONTROLS the precision of your DEMs so it needs to be clearly stated and rigorously examined.**

In contrast to the fore-mentioned Nikon D800 (where camera and GNSS are typically connected through the camera's flash shoe), the MACS camera itself merges the GNSS/INS data stream into the MACS image files in real time during the flight. Thus, the external georeferencing information is written into the metadata header of the corresponding image file. We have added more detailed information on the entire process in the updated Sect. 2.2 and updated Fig. 3 to include the information that the geopositional information comes from an external georeference.

Changes made: *see CC #12*
*Updated Fig. 3:*

[Figure]

Published outputs

**28. Please understand too that the Pix4D processing report gives no useful information on actual errors – it merely gives the MISFIT between the values you fed it and the values it determined in the bundle adjustment. You also must specify within Pix4D what you believe the accuracy of your photo centers is so that it won't go too crazy with adjusting them, and you should make clear in the paper (given all of the other uncertainties and problems using an opportunistic data set) what that value is given the novelty of using MACS for this purpose.**

The report of the Pix4D software contains the point "Absolute Camera Position and Orientation Uncertainties", which is according to the authors of the help documentation "similar to the expected GPS accuracy" Quality Report Help - PIX4Dmapper. The sensor heads of the MACS - Polar camera system were calibrated in the laboratory before the image flights. This resolved inner orientation is also introduced into the bundle block adjustment. Together with the accurate lever arms of the GNSS solution, it could therefore

be assumed that the improvement of the outer orientation must be within the expected accuracy of the navigation solution. This was assumed to be 1 m for the image flights, without post-processing of the navigation data, and introduced as a standard deviation. We added this information to Sect. 3.2.

Added in line 291, page 13: *"Within Pix4Dmapper, we specified the expected positional accuracy of the images to be 1 m."*

**29. Section 5.3 indicates that there are serious data quality issues here and I do not believe they are attributable to the causes given. Horizontal accuracy should be within 1-2 pixels, perhaps 50 cm at most, if this work is done to modern standards. I was mapping thousands of square kilometers at 10 cm ten years ago at 1-2 pixel accuracy and that is what scientists expect of data acquired since then (especially in 2021), so if you are getting 2-4 m horizontal mismatches then you need to make this very clear up front and determine how typical this is of the data you are providing. Vertical accuracy stated for this single project is poor but given the lack of ground control that's fine, the data are easily shifted vertically to match the lidar and in a sense for change detection the data could have no vertical reference and still be just as useful as long as common zero-change reference points are found and detrended in the comparison. What seems completely missing and ESSENTIAL is any discussion of vertical precision – these data were nominally collected and published for the purpose of change detection and the accuracy of change detection is described ONLY by the vertical precision of the individual data sets being compared. A rigorous assessment of vertical precision is required here and is done by DEM-differencing with a reference data set and examining the standard deviation or 95% RMSE of difference.**

As described in our response to CCs #25 and #26, we have now included a more in-depth accuracy assessment, including the comparison of our datasets to reference datasets such as the data provided by the NGS / NOAA / AHRI and DGPS measurements of GCPs.

Changes made: *see CC #25.*

**30. The authors mention somewhere that several blocks were acquired several times (perhaps at different AGL?) – these DEMs should be assessed for horizontal and vertical accuracy and precision too. Why would you not?**

This is correct, we have acquired the target "Cape Blossom" at the three different flight altitudes of 500 m, 1000 m, and 1500 m, which resulted in three datasets of overlapping extents at 7 cm, 10 cm, and 20 cm, respectively (see Table 2, target IDs 15, 16, and 14 respectively). We now have conducted an accuracy assessment for the horizontal and vertical accuracies and present the results in Sect. 5.1.

Changes made: *see CC #25.*

**31. If you want people to use these data in the future, you need to indicate what sorts of questions can be addressed by them! For example, can you use these data to detect permafrost thaw slumps before they occur or is it only the gross failures that**

**can be assessed? Can you use these data to assess ice wedge melt? Etc. Provide examples of this, like in Figure 13 but for cool stuff that actually worked well to excite and motivate readers to use your data.**

Thank you for your helpful suggestion. In the Conclusion, we've expanded the list of potential (scientific) research questions that may be addressed with our datasets and have added a figure to show some of these examples. We purposefully highlight features that strongly benefit from the high spatial resolution of our datasets, and that we have introduced earlier on in the manuscript (Introduction, Study areas).

We have deliberately not presented any more specific research results conducted with these datasets, as this is not within the scope of a data paper and explicitly not desired according to the ESSD guidelines "[...] extensive interpretations of data – i.e. detailed analysis as an author might report in a research article – remain outside the scope of this data journal". Any publications using these datasets so far and that we are aware of have been referenced.

Changes made on line 551, page 31: *"Potential research with these data sets may include, but is not limited to: tracking coastal erosion (e.g., impending block failures); the detailed analysis of ice-wedge polygons, their microtopography, and their degradation dynamics; monitoring thaw subsidence to evaluate potential impacts on infrastructure; the detection and characterization of retrogressive thaw slump and thermo-erosion gully dynamics; the detailed analysis of ground characteristics in recent and historic fire scars; the detection of lake drainages and drainage pathways in thermokarst lakes (Jones et al., 2023); the examination of individual shrubs and trees in the shrub-tundra regions; or the quantification of beaver dams and lodges (Fig. 17).*
*Given the absence of radiometric calibration targets during the campaigns, calculating radiometric indices may not be wholly reliable with the current version. However, it is important to note that these constraints do not diminish the datasets' utility for object-based analyses, automated segmentation tasks or the mapping of specific features, their distribution, and their microtopography."*

Added Fig. 17, page 33:

[Figure]

**32. Here are some of my papers and blogs which give a sense of what I mean by a rigorous accuracy and precision assessment for reference, each slightly different based on prior research and current topic. I'm not saying you need to do things my way (I chose my papers for my convenience), but you do need to leave the reader with a clear sense of the scientific questions that can be assessed with your data. You'll also notice that there are overlaps between some of our data sets that can be used for your data quality comparison.**

**Nolan, M., Larsen, C., and Sturm, M.: Mapping snow depth from manned aircraft on landscape scales at centimeter resolution using structure-from-motion photogrammetry, Cryosphere, 9, 1445–1463, https://doi.org/10.5194/tc-9-1445-2015, 2015.**

**Nolan, M. and DesLauriers, K., 2016. Which are the highest peaks in the US Arctic? Fodar settles the debate. The Cryosphere, 10(3), pp.1245-1257.**

**Swanson, D. K. and Nolan, M.: Growth of Retrogressive Thaw Slumps in the Noatak Valley, Alaska, 2010–2016, Measured by Airborne Photogrammetry, Remote Sens-basel, 10, 983, https://doi.org/10.3390/rs10070983, 2018.**

**Gibbs, A. E., Nolan, M., Richmond, B. M., Snyder, A. G., and Erikson, L. H.: Assessing patterns of annual change to permafrost bluffs along the North Slope coast of Alaska using high-resolution imagery and elevation models, Geomorphology, 336, 152–164, https://doi.org/10.1016/j.geomorph.2019.03.029, 2019.**

**https://fairbanksfodar.com/science-in-the-1002-area/**
**https://fairbanksfodar.com/fodar-makes-50-billion-measurements-of-snow-depth-in-arctic-alaska/**
**https://fairbanksfodar.com/the-first-fodar-map-of-denali-alaska/**
**https://fairbanksfodar.com/west-coast-village-data-delivered/**

Thank you for sharing your papers and blogs to provide insight into your perspective on rigorous accuracy and precision assessment. In our revised manuscript, we have incorporated references to some of your relevant papers in the Introduction and have added to the Discussion some of the scientific questions that can be assessed with our data (see CC #31). Unfortunately, we have not found any overlapping areas between our datasets described here and the ones that you have published. Thus, for the data quality assessment, we have chosen to compare our datasets with those published by the NGS (see also CC #25).

Changes made in line 61, page 3: *"[...] modern optical airborne datasets have been used to [...] coastal erosion (Jones et al., 2020; Obu et al., 2017; Gibbs et al., 2019), [...], retrogressive thaw slump development (Swanson and Nolan, 2018) and [...]"*

Changes made in line 551, page 31: *See comment #31*

**33. In section 4, labeled as describing data and metadata structure, there is a paragraph on GNSS accuracy (?!). This is the only mention that you had on board GNSS (which should be in methods) and the accuracies given here are exceptionally crude – 2 m vertically? How can this be? Using modern PPP processing, exclusive of blunders or poor system design, you should be achieving < 10 cm positioning and more like 1-2 cm. More detail needs to be provided on this in the method and processing sections (there is no information on GNSS processing or photo center geolocation methods that I could find). Mention is made here that the photogrammetric data are going to be reprocessed once the GNSS data are reprocessed – why then are you publishing this paper and these data now? Don't you think this will simply add confusion by publishing multiple versions? GNSS processing, even when tightly coupled to IMU, takes only a few hours and it seems these are small blocks that only take a few hours each to process photogrammetric, so I think this should be done before this paper is published, along with a rigorous accuracy assessment.**

We agree that by including the GNSS correction prior to photogrammetric processing, the accuracy of the datasets would improve greatly. Thus, as detailed in the response to CC #25, we have decided to go back to square one and reprocess all datasets with the GNSS-corrected navigational data, which is now available to us. With this correction, we now achieve positional accuracies in the centimeter-scale. We have included the information of GNSS post-processing in Section 3.1.1 and the results of the quantitative accuracy assessment in Section 5.1. The newly processed datasets are currently being handled by PANGAEA to become freely and openly accessible (with DOI) to any interested user. However, since this process takes PANGAEA multiple weeks to months, we have uploaded a representative subset to a Google Drive:
https://drive.google.com/drive/folders/1hBJc5lYjvbDdIuu18Z1IJYt3LBIuT4ct?usp=sharing.
This link is for review purposes only and will not be maintained once the upload of the reprocessed datasets has been finalized via PANGAEA.

Changes made: *see CC #25.*
Added in line 172, page 7: *"The sensors of the MACS camera system were calibrated in-flight. During the campaign in 2021 topology points of an urban scene in Kotzebue were measured with a double frequency GNSS receiver (Leica VIVA GS14). In a GNSS post processing with open IGS data (https://igs.org/) the points were calculated in sub-centimeter accuracy. Together with MACS aerial images from the same scene, we applied a bundle adjustment and derived the interior orientation camera parameters. These camera parameters were then used for further image processing."*

**Science/technical questions**
**34. Reference is made to the total area covered by your data. How were these areas calculated? This data purports to present only the blocks that were processed photogrammetrically. I have mapped blocks that are 6000 km2 and there is no way that your small blocks add up to anything near this amount as is stated in your conclusions, so I believe this is misleading and potentially disingenuous. In this paper you should limit your discussion only to the blocks you are presenting and I**

**think that will help focus the paper in many ways overall, though a single panel figure (like 1d) is fine to set the context.**

We have reviewed the coverage of our published datasets and report that the blocks actually only cover a total of 1591.32 km². We believe that our mistake came from a simple slip in the correct column of a table. It was certainly not our intention to mislead readers. We apologize for the confusion.

Changes made in line 544, page 30: *"[...] covering more than 1500 km² in [...]"*

**35. Great words are used to describe the MACS camera, but the results don't seem that impressive to me. These words should be toned down and a discussion made comparing to modern prosumer cameras which seem far superior to me based on your results. I understand completely that this project may be stuck with the data it has and that this is perhaps an opportunistic project based on those data – that is all fine, but be clear about this. If you are not proposing that everyone should use a MACS camera, then be clear about that. Just because it was a great thing 10 years ago and now is outdated doesn't mean that's bad, just be clear and honest about it.**

Thank you for your feedback. We believe our description of the MACS camera is neutral and not intended to promote its use over other options. We have no commercial interest in the development of the MACS and its datasets, and the MACS will also not become a widely-available system on the remote sensing market, as its development serves specific scientific and engineering purposes. We have provided a few details on the system, as this is one of the first publications that deals with data acquired from it. However, our primary goal is and has always been for ESSD (which is not a remote sensing or technology-oriented journal) to publish the dataset and provide a comprehensive description of its acquisition. Within the text, we have now replaced some instances of the word "MACS" with the words "system", "sensor", or "camera" to decrease the impression that we are promoting the system we used.

Changes made in:
- changed three instanced from *"MACS" to "camera system" / "sensors"*
- changed two instances from *"MACS images" to "raw images"*
- removed five instances of *"MACS" / "MACS-derived"*

**36. Why did you use Pix4D rather than other options? It's fine that you did, but there are other (probably better) options like Metashape – why did you not use that? If Pix4D was your only option for whatever reason, that's fine – just be clear.**

The choice of Pix4Dmapper was informed based on available experience and recommendation from DLR. It is important to note that while Pix4D was selected, it does not imply that any other software would not have been equally suitable for our specific requirements. While we acknowledge that Agisoft Metashape is a comparable and potentially equally suitable software on the market, we understand that Agisoft LLC is a Russian-owned and Russia-based company from St. Petersburg. Due to the current geopolitical situation and the potential to infringe on EU or US sanctions we therefore refrained from using this software.

Added footnote on page 10: *"We selected this software based on our expertise and availability."*

**37. Also be clear that everything needed for someone to reprocess the data on their own is provided, if that is indeed the case.**

Yes, this is very helpful for a reader/user to know! We have added a sentence ensuring this to Sec. 7 Data availability.

Changes made in line 588, page 34: *"This repository contains all necessary files and code to reproduce our data processing workflow exactly as described in this publication, assuming they have access to Pix4Dmapper (our workflow was only tested on v.4.6.4). The raw images can be provided upon request."*

**38. The data are described as multispectral and some discussion occurs on radiometric scaling, but I didn't understand it and my gut says that it is a bit unfair to describe these data as multispectral if that word is to retain any useful meaning. I mean RGB is technically multispectral but we don't refer to it as such. I didn't understand section 3.1.2 at all so this section should be cleaned up. And without radiometric calibration on the ground or some other means, again I'm not sure you're making a good case or instilling confidence in your readers for calling the system multispectral.**

In remote sensing, the term "multispectral" refers to systems with three to ca. 10-15 spectral bands. Given that we have four bands (B, G, R, and NIR), we believe that it is okay to refer to the system as multispectral. In line 66, page 3 we have specified "The data includes very-high-resolution multispectral images in the visible (red-green-blue, RGB) and near-infrared (NIR) wavelengths, [...]". With these specifications, we are quite explicit what bands we use and therefore what "multispectral" means in terms of our study. However, we understand that the term "multispectral" has been repeated quite often in the manuscript, maybe leaving the reader with a feeling of heightened importance of this term. As there is no reason to specifically emphasize this, we have removed the adjective "multispectral" in some instances or simply replaced it with "BGRN", making it unambiguous which bands we are referring to. We have also revised Section 3.1.2 to be clearer about our processing steps (and included an example).

Changes made:
- removed seven instance of *"multispectral"*
- changed one instance of *"multispectral"* to *"BGRN"*

Changes made in line 268, page 13: *"[...] For radiometric matching between the RGB and the NIR acquisitions, we applied a linear scaling factor to the RGB data. We determined the scaling factor from the difference in shutter timing between the two sensors. For example, if, during the flight, the shutter timing for the NIR sensor was set to 0.2 ms, and for the RGB sensor to 0.4 ms, the scaling factor here in the post-processing equaled 0.5. Subsequently, we scaled the data values between 0 and 65535 to exploit the entire 16-bit information range. This step results in a more homogenized output among the three campaigns and the*

*many target sites that were flown under varying illumination conditions and with slightly different initial camera parameter settings. [...]"*

**39. By 'shutter timing' did you mean 'shutter speed'? If so, why are your RGB cameras not using the same shutter speed? And how are you ensuring that they were acquired simultaneously?**

Thank you for pointing out the confusing term we have built in here. Since our camera does not use a mechanical shutter, we should rather use the term "integration time", which refers to the total exposure time of a sensor for each image. All sensors (two RGB sensors and one NIR sensor) acquire at the exact same time; this is ensured as they are triggered electrically on the software side. As we have revised Sec. 2.2, this information has been included there as well.

Changes made on line 164, page 7: *"All sensors are electrically triggered to start the image exposure at the exact same time. At the end of integration, the sensor delivers a pulse to the GNSS receiver, generating a message including information on the time, position and attitude of the acquisition."*

**40. Did you really provide Pix4D with O,P,K or was it actually yaw, pitch, roll? Just double checking.**

Yes, we did provide Pix4D with Omega, Phi, Kappa. But MACS metadata provides both: O, P, K, and yaw, pitch, roll. The yaw, pitch, roll information would therefore be available if other software has this requirement.

**41. What is the value of combining the point clouds for RGB and nIR in making gridded elevation models? Clearly they are measuring slightly different things and different contrast features – are you making an argument that this will lead to improved results? What analyses can you provide that back that up? You mention in Section 3.3 that it yielded the "best" results but give no indication of how you determined this.**

In order to determine from which sensor we should source the data to generate the point clouds for subsequent DSM generation, we selected three small study areas and conducted a qualitative analysis. For each of the study areas, we generated three DSMs: One from only the images collected with the NIR sensor, one from only the images collected with the RGB sensor, and one from all the images collected by both sensors. From a manual, qualitative assessment, we determined that for most areas, the 'combination-DSM' contained fewest errors, especially around sharper edges, such as buildings or thaw slump headwalls. The source of the errors could almost always be ascribed to over- or undersaturated pixels. Undersaturation often occurred in the RGB images (shadows of steep headwalls), oversaturation more in the NIR images (reflective materials of rooftops). This analysis has been described in the Appendix and referenced in Section 3.3 of the original manuscript.
In terms of the published data, we provide both the RGB point cloud as well as the NIR point cloud individually. We do this, as different users have different scientific goals and thus requirements to their data. For some users, the NIR point cloud may be more appropriate, for others the RGB point cloud.

Changes made in line 592, page 34: Rewrote some parts of the section:

*"To generate photogrammetric DSMs (as described in Sec. 3.3), we had three possible sources to select from: (a) The point clouds derived from the RGB images, (b) the point clouds derived from the NIR images, or (c) the combination point clouds from RGB and NIR. To determine which of these sources generated the best DSM results, we conducted a small test. For this, we selected three exemplary datasets that cover different landscape types and permafrost features represented over the entire available image space. The first AOI is found around the Selawik Thaw Slump in West Alaska. This thaw slump shows steep edges, as well as individually standing trees. The second AOI is Kivalina on Seward Peninsula, West Alaska, representing a village with buildings of different sizes. Finally, the third AOI is near Teshekpuk Lake. Here we find small elevation differences between the ice-wedge polygons and their troughs in between. It is important that the selected source allows the DSM-generating algorithm to both preserve the fine elevation details, and correctly represent steep or sharp edges, such as from buildings in the villages, a thaw slump's head wall, or of the individual trees.*

*Our analyses show that there is no general tendency towards either the NIR or the RGB sensor being the better option. Rather, the matching algorithm performs badly for oversaturated or undersaturated and very dark pixels in the original image. Within our three comparison AOIs, this effect can be seen in the DSMs from the RGB-only point clouds for Kivalina and the polygonal tundra near Teshekpuk Lake. In Kivalina, many metal roofs of buildings show oversaturation in the images and thus complicate the correct matching of pixels. This results in frayed and imprecise building edges. Similarly, the undersaturated water areas from the thermokarst ponds in the polygonal tundra AOI also show imprecise matching in the RGB-only DSM (Fig. A1i). As oversaturation can also be a problem in some NIR images (see Fig. 13), we found targets where the NIR-only point cloud is also affected by this issue. Furthermore, using the NIR-only point cloud, we also observed that the resulting DSMs showed less sharp edges in comparison to the DSMs from the RGB-only point clouds. This is a result of the lower point density of the NIR point clouds. This effect can be seen both at the thaw slumps head wall edge in Figs. A1d and g and the edges of buildings in Figs. A1e and h. Using the combination point cloud (Fig. A1k-m) can overcome the worst of both the described effects and results in the most coherent DSMs for the majority of our targets.*

*Within our published datasets, we provide both the RGB and the NIR point clouds, should users desire to reprocess a certain DSM with only one of the point clouds."*

**42. In Figures 7-9 you show data examples, but the location map seems to indicate enormous areas covered in these blocks (presumably that's what the red area is on the location map?) which is not what Figure 1 shows. Could you clarify?**

We understand that the overview maps in Figs. 7-9 might have been misleading. The red squares simply showed the general location of the example dataset in panels a-e. They were not intended to be interpreted as the footprint. We have changed the symbol from the square to a star to make this unambiguous for all readers.

Changes made: Updated overview maps for Figs. 7-9.

[Figure]

[Figure]

[Figure]

**43. In Section 5.2 you mention cloud cover requiring 'longer sensor exposure' – do you mean shutter speed here? Is the MACS system not capable of adjusting ISO? Can you clarify this? Also can you specify what range of shutter speeds you used and the speed of the airplane and the associated percentage of pixel blur while the shutter was open?**

In consumer and prosumer cameras the term "ISO" is used to control gain parameters of the internal A/D converters. This factor controls the "brightness" of an image, but generally speaking, does not add any image information, nor does it make the sensor more "sensible" in low light conditions. Thus, in the field of industrial cameras and professional remote sensing, there is no ISO equivalent. With our camera system being an industrial sensor, it therefore does not require ISO adjustments. We have added information on the exposure times, the airplane speed, and the motion-induced blur in Sec. 2.3 Survey design (see also comment #16).

Changes made in line 200, page 9: *"For our aerial camera all parameters are fixed except exposure time, typically ranging from 0.2 ms to 1.5 ms. Given a motion rate of 6.7 cm/ms (aircraft speed was roughly 130 kts), the resulting motion-induced blur at e.g., 1.5 ms exposure time is approximately 1.2 pixels for an 8 cm GSD scenario. During some of the flights, we chose different shutter timings for the NIR and the RGB sensors, to accommodate for the specific environmental conditions, such as brightness levels of the ground features, or expected shifts in illumination throughout the survey time."*

**44. Here you also mention HDR techniques but I did not understand it. Can you clarify? Are you attempting to merge several photos together? That's what HDR normally means. Are you taking two photos at each intended location but with different shutter speeds? How exactly were these multiple photos used and how does this affect DEM accuracy and precision compared to using a single photo? Or did you just use a single photo (which is then not HDR)? Does this mean that you had no ability to change shutter speed in flight?**

Thank you for pointing out this issue. We acknowledge that the term HDR in the context of our camera is incorrect. Colloquially and internally, we have been using "HDR" to refer to the fact that the camera can capture multiple different exposure times simultaneously. Of course, this is not what "HDR" refers to in the technical sense in photography. We have thus removed the term "HDR" and more accurately described the camera's capability of capturing imagery with different exposure times.

- Added in line 227, page 10: *"For targets where we expected to see both very bright and very dark ground features (e.g., snow and water bodies), we also acquired images at two different exposure times (e.g., 0.4 ms and 1.0 ms), to ensure that we*

*always had at least one image that was not over- or underexposed. This was also done when we expected shifts in illumination throughout the survey time."*

- Added in line 294, page 13: *"For flights, where we acquired images at multiple different exposure timings (e.g., 0.4 ms and 1.0 ms, see Sec. 2.3), we input both image sets into Pix4Dmapper, and let the software select the adequate image information for processing."*
- Removed in line 403, page 23 (numbers from old manuscript): *"For flights during the later campaign in West Alaska in 2021, we captured imagery using multi-exposure (usually at 1 ms and 0.4 ms) whenever inconsistent cloud cover seemed probable. With this setting, we were able to decide later which of the two exposure times generated the better image for each individual target area and only considered those for further processing."*

**45. In section 5.3 you describe acquiring the TVC in race track format rather than flying adjacent flight lines in grid sequence. Having tried this myself occasionally, I can tell you that my conclusion is not that changing illumination (that is clouds or something) but rather the sun angle causes the increased errors. Even though there is not much vegetation here, the primary contrast features picked by the photogrammetric software are shadows, and over a 3 hour acquisition the shadow direction is changing 45 degrees in the Arctic. So it's always best, from what I found, to minimize the time between adjacent flight lines for this reason and only use the race track approach when logistics call for it. For example, if you are mapping a road or field site and it looks like the weather won't hold for the entire time you need, map the highest priority location in the center first so you're sure you get it then expand in a racetrack format until the weather finally calls the show. Otherwise if you start at one side of a block and fly in a normal grid sequence, you may not reach the most important area before the weather shuts you down. Same thing but worse if you spiral in on your highest priority from the outside.**

Thank you for these suggestions, we certainly agree! Throughout multiple campaigns, we have explored different flight patterns and have found advantages and disadvantages for race-track, spirals, and consecutive lines. As you know, flight planning is informed by the target, the terrain, the expected weather, and the expected illumination, among others, and spontaneously changing these flight patterns - which sometimes becomes necessary only when the aircraft is already airborne - is not always feasible, in particular under the additional logistical constraints of working in the Arctic. For the past campaigns, we can unfortunately not correct the flight planning choices, so we try to get the most out of the data we have. For future campaigns, we will certainly consider the race-track approach starting with the center as you suggest.

In terms of the cause of the visible flight lines (as described e.g., in Sec. 5.2.3), we agree that this is due to the changing illumination angle over time, and have added another section on this (see also comment #16 of RC2).

Changes made line 511, page 29:
*"5.2.3 Illumination angle and bidirectional reflectance distribution function*
*A second source for visible linear artifacts stems from the bidirectional reflectance distribution function (BRDF). The BRDF describes how surfaces reflect light at different*

*angles of incidence and reflection. Therefore, when a flightline is directed towards the sun and the neighboring line is flown in the opposite direction (away from the sun), this variation leads to changes in the perceived reflectance of the surface. Surfaces with different BRDF characteristics will reflect light differently based on the flightline angle. This variation affects the radiometric properties of the captured imagery, causing variations in brightness, contrast, and spectral response across the images (examples can be seen in Figs. 7 and 14a).*

*Photogrammetric matching is additionally impeded by larger time gaps of neighboring flight lines. Processing software will often select high-contrast features to prepare for image matching, which may include shadows cast on the ground. The more time that has passed between two flightlines that should be matched photogrammetrically, the larger the induced error however, as any shadows wander across the ground with a changing sun illumination angle.*

*The safest way to avoid such artifacts is by already factoring in the sun position into the flight planning phase. In our case, this was often not possible: Preparing flight plans for airborne surveys is a lengthy process and thus needed to be done ahead of the campaign. However, the decision which targets would be flown on a given day was only made each morning, based on the local weather conditions at the desired target sites. Thus, a spontaneous realignment of the flight direction according to the sun position was not feasible on such short notice. Some separate post-processing techniques to mitigate such artifacts have been proposed by i.e., Queally et al. (2022); Greenberg et al. (2022); Wang and Liu (2016), but have not been tested on the MACS datasets. Depending on a user's requirement and their desired application, some algorithms might be more suitable than others."*

**46. Section 5.4 on water areas does not match my experiences. The claim is made here, I think, that white caps are usable photogrammetric features. If the goal is just to get any topographic result so that an orthoimage can be made that may be true. But the photogrammetric bundle block adjustment depends on the observed parallax in contrast features to be solely due to topography – if the contrast features are moving (like shadows, waves, cars, etc) then the topographic measurement will be thrown off. It seems that this is recognized here, but it is not clear why the topic is addressed and additional clarity would be useful if I am missing something.**

Thank you for this summary! After reviewing our initial paragraph, we acknowledge that it can be interpreted in a misleading way. In general, we are on the same page, but the paragraph did not properly reflect that. We have rephrased the section in question and hope that the overall message is now clearer.

Changes made in line 537, page 30: *"In contrast, when we encounter waves or whitecaps in the ocean or in wind-blown lakes and rivers, the color contrast does allow matching during the automated SfM processing. Therefore, these areas do generate sufficient points within the dense point cloud to interpolate when creating the DSMs with WhiteboxTools. However, this is not necessarily a desired effect, as such white caps are moving objects between the pictures taken. Thus, any topographic information derived from these areas is likely false and should not be considered in any analysis. We see this effect for example in the dataset of Shishmaref, a village on an island in the Chukchi Sea (Fig. 16)."*

**47. Reference is made in several places that these data will be useful as training data for machine learning use in satellite-based studies but no mention I could find was given as to how or for what scientific purposes. These comments should either be removed or described in more detail, especially in reference to specific blocks in this dataset and presumably especially those that repeated prior mapping.**

As addressed in comments #20 and #31, we have elaborated on the potential scientific purposes that our data may be useful for in the Discussion. We have also added another column to Table A1 that includes tags of features to be found in the respective datasets

Changes made: *See comments #20, #31.*